# Risk caused by the propagation of earthquake losses through the economy

J. A. León [1✉], M. Ordaz [1], E. Haddad [2,3,4] & I. F. Araújo[2,4]

The economy of a country is exposed to disruptions caused by natural and man-made disasters. Here we present a set of probabilistic risk indicators, the Average Annual Loss (AAL) and the Loss Exceedance Curve (LEC), regarding to production, employment, Gross Domestic Product (GDP), Gross Regional Product (GRP), export volume, inflation, tariff revenue, among others, due to earthquakes. All indicators are computed using a systematic probabilistic approach, which integrates the seismic risk assessment with spatial computable general equilibrium models, both robust and well-known frameworks used worldwide in their respective fields. Our approach considers the induced damage and frequency of occurrence of a vast collection of events that collectively describe the entire seismic hazard of a country, giving us a better and more complete understanding of the full consequence of earthquakes. We illustrate this approach with an example developed for Chile.

[1] Instituto de Ingeniería, National Autonomous University of Mexico (UNAM), Ciudad Universitaria, Coyoacán, 04510 Mexico, DF, Mexico. [2] Department of Economics, University of Sao Paulo, Av. Prof. Luciano Gualberto, 908, FEA 1, Cidade Universitária, Sao Paulo, SP, Brazil. [3] Faculté de Gouvernance, Sciences Économiques et Sociales, Université Mohammed VI Polytechnique, Rabat, Morocco. [4] The University of Sao Paulo Regional and Urban Economics Lab (NEREUS), Av. Prof. Luciano Gualberto, 908, FEA I,Cidade Universitária, Sao Paulo, SP, Brazil. ✉email: jleont@iingen.unam.mx

There is evidence that economic production losses caused by disasters, on occasion called indirect losses, can sometimes be much more significant than those produced by physical damage[1,2]. We have witnessed recent catastrophic events that have made evident the economic implications and new features of disaster. The 2004 Indian Ocean earthquake and tsunami, hurricane Katrina in 2005, the 2010 Haiti earthquake, and the 2011 Japan earthquake and tsunami highlighted the importance of matters as disaster risk governance, reconstruction strategies, the vulnerability of the economy of developing countries, the multinational disaster aspects and how disasters can cascade. Accounting for indirect losses in risk is crucial and becomes more prominent as the supply-chain complexity increases in the age of globalization[3–6]. However, it is not easy to account for the economic consequences of earthquakes with historical information alone because these catastrophic events are infrequent, so that relevant information is scarce, and it is not always easy to distinguish between direct and indirect losses.

Catastrophe modeling of physical damage caused by earthquakes is today a well-developed technique, to the point that it is possible to estimate, in a probabilistic manner, the seismic risk of individual assets[7–10] (infrastructure, buildings, contents, machinery, equipment, etc.) Modeling of indirect losses, in comparison, lags far behind, mainly due to the difficulties in empirically translating property damage into indirect losses[5] and due to the lack of adequate models that relate these two kinds of losses. Usually, indirect losses are estimated roughly as a percentage of physical losses, establishing this percent with empirical information obtained from a minimal historical database of events worldwide; this procedure is deficient and with very low reliability.

However, there is a well-developed literature on the economic impact of natural and man-made disasters, whose most recent advances have been compiled by Okuyama and Rose[11]. On the one hand, efforts have been focused on improving and extending the quantitative models used for disaster impact analysis, such as cyber-attacks, extreme weather events, earthquakes, flooding, climate change, and terrorist attacks, amongst others[12–17]. In addition, new frameworks have been proposed to integrate models of transportation, critical supply chains, and community demand[18]. On the other hand, researchers have devised tools to directly estimate economic damages or losses based on some physical data of natural hazards[19,20], or to evaluate, for instance, the changes in economic activities with a set of satellite data on annual differences in nighttime light intensity[21]. Furthermore, ESPON-TITAN[22,23] provides evidence on the direct and indirect economic impacts of natural hazards, extreme events, and disasters, identifying trends and territorial vulnerability patterns across European regions and cities. This research project uses multi-regional input-output models to assess the costs of extreme events on the supply chain. The European Commission[24] and the World Bank[25] also use the input-output models to assess direct and indirect economic impacts of disasters in Europe.

Despite this large and rich analytical framework to study the impact of disasters on the economy, efforts have concentrated on analyzing individual events[26–30], without appropriate consideration of their frequency of occurrence. But the frequency of occurrence matters. Decisions and policy would be very different if one knew that the economic impacts of a specific size are to be expected, on average, once every 100 years or once every 1000 years. Further, it would seem that little attention has been given to linking physical damage to infrastructure or economic components with higher-order economic losses.

Given this, here we present an approach to integrate, systematically, the probabilistic seismic risk[31–35] and the Computable General Equilibrium[36–40] (CGE) modeling frameworks to the

estimation of higher-order losses that take into account that: natural events—earthquakes in our case—take place as a stochastic process in time; that frequency of occurrence of events matters for measuring risk; and that there are links between the level of physical damage to economic components and the reduction of capital stock. We use a particular type of CGE model, known as spatial CGE models[41,42], which can consider the geographic location of economic agents and endowments. The "Methods" section presents in detail the methodology proposed for the integration of the models. Besides successfully dealing with the estimation of indirect monetary losses, our approach allows for a better glance at the likely consequences of the earthquake occurrence in the whole economy by carrying out risk assessment for a vast set of events (44,350 events for our case of study) and referred to different economic variables, such as employment, GDP, GRP, wages, tariff revenue, consumer price index (CPI), export volume, among others. In our hand-on case of study, we computed the annual average loss and loss exceedance curves for several components of the Chilean economy at the country, regional and sectoral level. The LECs allow us to estimate the expected losses associated with a wide range of return periods. Furthermore, we have observed the capacity of our model to catch positive economic impacts of earthquakes on specific sectors and regions through substitution effects. We believe that the results and the approach presented add value to current knowledge for disaster risk management, providing a set of complementary risk indicators in catastrophe modeling.

## Results
As we have mentioned before, this work proposes a robust, probabilistic, and systematic connection between entire seismic risk and large-scale economic models. The relation between the reductions of the capital stock and the direct losses in non-residential buildings are estimated from the probabilistic model for seismic risk assessment. The results obtained in this example are pretty reasonable and can be used to gauge the power of the approach. In what follows, we will present the main results of our example, highlighting some of the most interesting findings.

**Overall risk results.** Due to direct or indirect losses, we will express seismic risk results in terms of a few standard risk metrics. We will use the average annual loss (Methods, Eqs. 1–2) and the loss exceedance curve (LEC; Methods, Eqs. 3–4). Let us start presenting the results at the country level.

The more aggregated results of our analysis, in terms of the AAL, are presented in Table 1. As shown, the direct AAL for the whole country has been estimated as 302 million dollars while the AAL of production losses reaches 0.132% of the total yearly production of the country. Direct losses computed refer exclusively to non-residential buildings. This restriction is that we are trying to model damages only in capital stocks used in the production process, in the sense that their physical disruption is susceptible to being propagated in the country's economy.

The LEC for physical losses is a standard metric within the risk assessment world; however, the LEC for production losses is introduced in this paper (Fig. 1a). Now, we have the means to estimate the production loss expected for any return period. For instance, the production loss associated with 250 years of return period for Chile was estimated at 15,870 million dollars (3.58% of the total yearly production) while direct losses were 5025 million dollars, that is 4.85% of the total value of non-residential buildings. For 1000 years of return period, the production and direct losses were 28,760 and 9835 million dollars, respectively. Note that in catastrophic risk modeling, it is customary to indicate the likelihood of an event taking place in, say, the

**Table 1 Aggregated results of losses at the national level.**

| Item | Average annual loss | Total exposed value (2014) | AAL as % of the total value |
| --- | --- | --- | --- |
| Non-residential buildings | 302 million USD | 103,720 million USD | 0.290 |
| Yearly production | 583 million USD | 442,805 million USD | 0.132 |
| GDP | 305 million USD | 251,020 million USD | 0.122 |
| Employment in Chile | 7786 workers | 6,751,073 workers | 0.115 |
| Export volume | 62 million USD | 83,102 million USD | 0.075 |

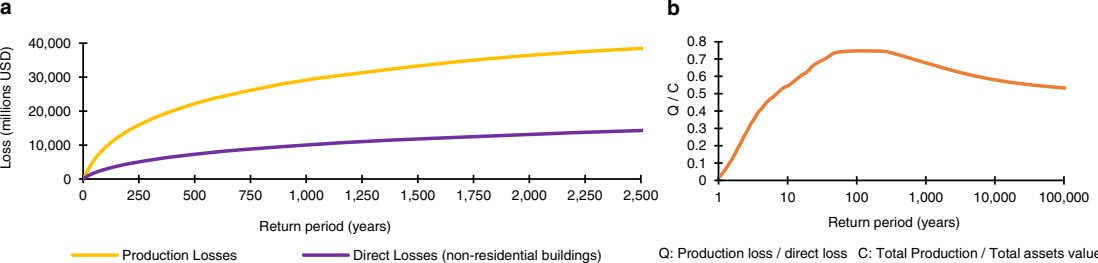

**Fig. 1 Loss Exceedance Curves (LECs) of direct and production losses.** We are representing the LEC with the return period (the inverse of the exceedance rate) in the horizontal axis and the loss values in the vertical axis. **a** Shows the LECs expressed in million dollars. Direct losses refer exclusively to non-residential buildings of the country. **b** Shows the relation between production losses and direct losses, $Q$, as a function of the return period. The ratios are presented normalized to $C$, being $C$ the ratio between total production and the total value of assets (non-residential buildings).

following year by using its *return period*. (See Supplementary Note 1 in Supplementary Material).

We found that for less severe events with low return periods (up to 50 years), the production losses follow a positive proportional relation with the direct losses, which goes from almost zero up to a maximum value of $0.74C$, $C$ being the ratio between total production and the total value of non-residential buildings (Fig. 1b). We observed that production losses are larger than direct losses for losses with return periods between 50 and 400 years. In a third stretch, as soon as the losses become less probable and more severe (return periods higher than 400 years), the production losses present a very soft negative proportional relation with direct losses. For instance, for return periods of 10,000 and 100,000 years, the relation between production and direct losses dropped to $0.6C$ and $0.5C$, respectively.

Risk metrics can be computed by economic sector or region of the country (Figs. 2, 3 and 4). Our results reveal that both the AAL distribution by sector and the risk rank of economic sectors are different for direct and indirect losses (Fig. 2). For instance, in the case of direct losses, the riskiest sector and the one that contributes the most to its total AAL, with 35%, is sector S6 "Commerce, hotels and restaurants". However, regarding production losses, the riskiest sector is S7 "Transport, communications and information services" while the one with the highest contribution to its total AAL is S3 "Manufacturing industry", with 23%.

We can also see that the regional distribution of the AAL of direct losses of Chile is similar to the regional distribution of the AAL of production losses (Fig. 3 and Supplementary Fig. 1). As expected, the Metropolitan Region of Santiago, R7, concentrates the most significant part of the average annual loss: 40% of the AAL of direct losses and 41% of the AAL of production losses. On the contrary, regional risk indicators of direct losses are not proportional to their corresponding ones of production loss. We have found that the riskiest region (highest value of relative AAL) in terms of direct losses is the Region of Atacama (R4), while in terms of production losses, the riskiest is the Region of Valparaiso (R6). In the case of the Region of Antofagasta (R3), it is the 5th riskiest region in terms of direct losses and the 12th in terms of

production losses. This demonstrates the importance of considering the high-order economic losses since, while there are regions most physically affected by the earthquake hazard, there are other regions that suffer the largest effects in terms of loss of production.

As mentioned before, our approach also allows us to obtain the LEC of production for whole regions or economic sectors of the country (Fig. 4), being able to compute the loss in a region or sector for any return period. For example, for a return period of 250 years, region R7 shows a loss of 10,674 million dollars, 5.45% of its total yearly production. However, in relative terms to its corresponding regional yearly production, up to a return period of about 450 years, the most affected region is R6, while for return periods greater than 450 years, the most affected one is R1 (Supplementary Fig. 2). In the case of economic sectors, the manufacturing industry, S3, presents the highest losses in the country in absolute terms, and, transport, communications, and information services, S7, the highest losses in relative terms to its sectoral production for any return period.

In addition to results aggregated at the national level, we can analyze what happens within the regional economics of the country by computing both the AAL of production losses and the production LEC of economic sectors into a specific region, as shown in Supplementary Fig. 3 and Supplementary Fig.4. Clearly, those figures show the differences in the production loss behavior of economic sectors in each region of Chile. This individual risk characterization allows us to study a particular region considering the economic interactions generated at the whole country, with the foreign component modeled as a single agent.

**Complementary risk indicators**. The richness of CGE modeling regarding the number of interesting output variables, each reflecting a different aspect of the economy, allows for the computation of many interesting and valuable risk measures. Besides calculating the risk measures in terms of production loss, it is possible to compute risk measures for losses of employment, GDP, GRP, tariff revenue, CPI, export volume, among others. These complementary indicators can be helpful for seismic risk

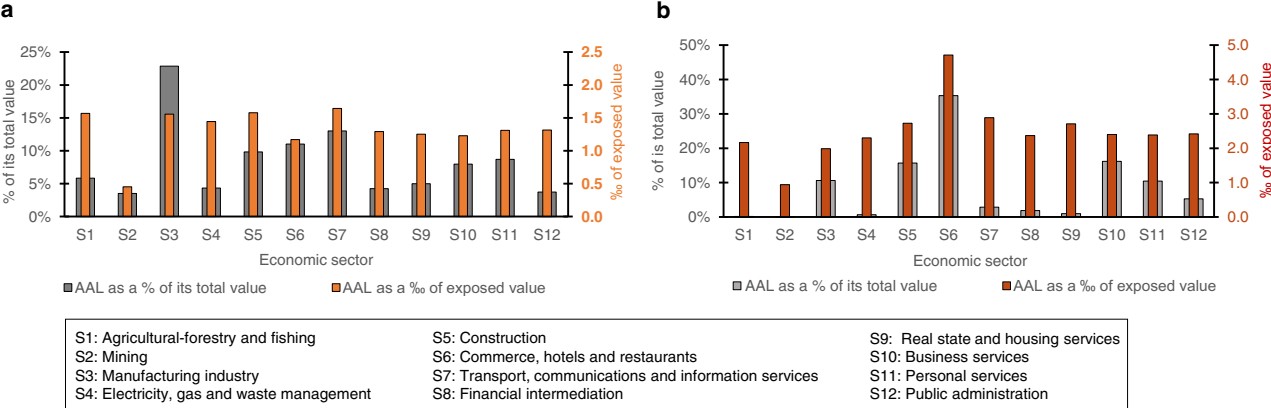

**Fig. 2 Average Annual Loss (AAL) for Chile by economic sector. a** Shows the production AAL as a percentage of its total value (gray) and as a fraction (per thousand) of its yearly sectoral production (orange). **b** Shows the direct AAL as a percentage of its total value (gray) and as a fraction (per thousand) of the corresponding sectorial value of non-residential buildings (red). AAL is presented as a percentage of its corresponding total value to see the influence of each sector in the total AAL and as a fraction (per thousand) of its corresponding exposed value to see how risky each sector is.

**Fig. 3 Distribution of Average Annual Loss (AAL) in Chile. a** Shows the seismic hazard for Chile, expressed in terms of the Peak Ground Acceleration (PGA) associated with 475 years of return period. The average annual loss due to physical damage of non-residential buildings (direct losses) by region is shown in (**b**) in millions of USD and (**c**) as a fraction (per thousand) of its regional exposed value. The average annual loss in production by region is shown in (**d**) in millions of USD and (**e**) as a fraction (per thousand) of the corresponding regional annual production. Finally, (**f**, **g**) present the Annual Average Gain (AAG) of production by region, due to substitution effects, in millions of USD and as a fraction (per thousand) of the corresponding regional annual production. AAL is presented as a fraction (per thousand) of its corresponding exposed value to see how risky each region is.

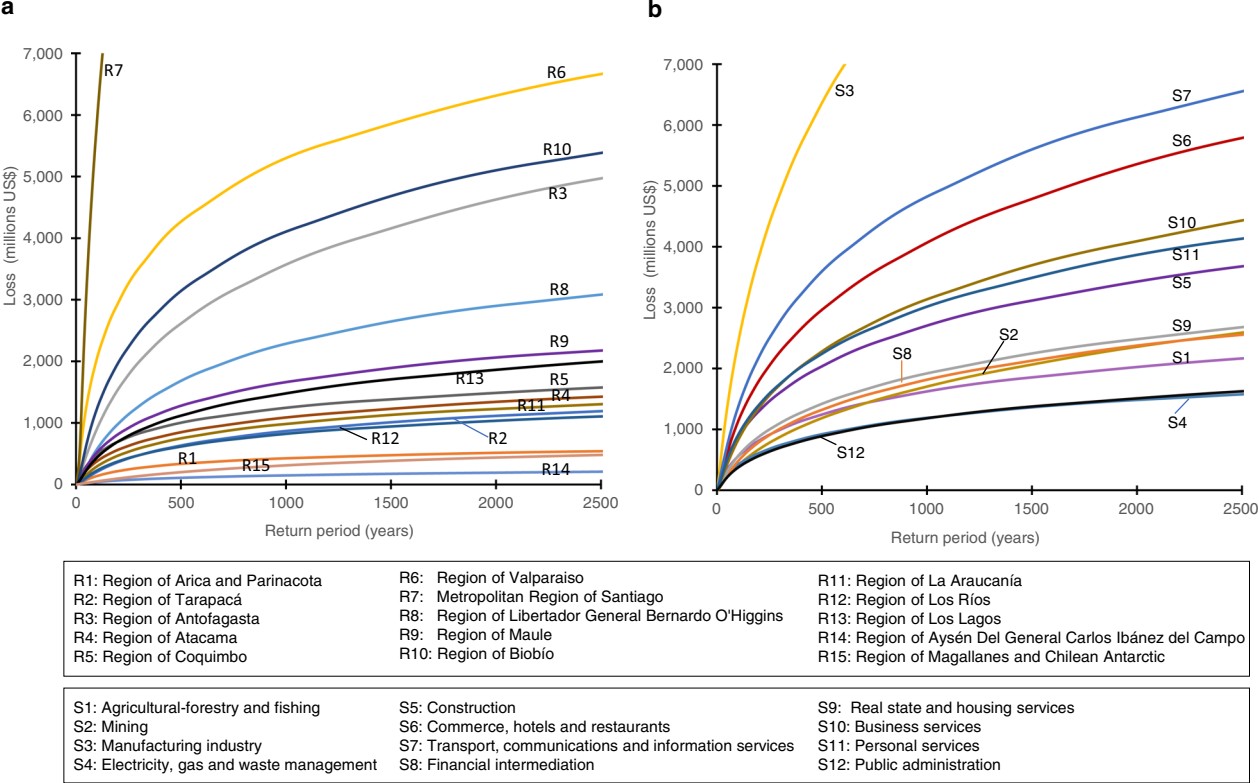

**Fig. 4 Loss Exceedance Curves (LECs) of Chilean production by region. a** By region of the country, (**b**) by economic sector. Note that LEC is presented with the return period (the inverse of the exceedance rate) in the horizontal axis and the loss values in the vertical axis.

management, as they provide a way to measure the losses in various aspects of an economy facing earthquake hazards.

For instance, as presented in Table 1, we have computed the AAL of employment, GDP, and export volume for Chile in 7786 workers, 305 million (0.122% GDP contraction), and 62 dollars, respectively. In addition, we present the LEC of employment, GDP, and export volume for the whole country in Fig. 5. Results can also be explored at the regional level precisely in the same way for production losses, as shown in Fig. 6, where we present the regional AAL for different economic variables (see also Supplementary Fig. 5 in Supplementary Material). In the case of Chile, the Metropolitan region of Santiago is the one that loses more employment, on average, with 4520/year. However, the most affected region concerning its total employment is Valparaiso (R6), with an AAL of 0.15% of its total employment. LEC allows us to make the economic estimation of risk for a wide range of return periods. For instance, the export volume loss in Chile associated to 250 years return period was 2.25% of its total yearly export volume, while Chilean's GDP was 0.122%. Other economic variables like the CPI that more than a quantitative measure can be an example of a qualitative indicator of risk that can be used to see the effect of earthquakes on inflation either at the country level or at the regional level (Fig. 6g). We see that regions located at the center-north of the country are more sensitive to the price increase due to earthquakes (see also Supplementary Fig. 6a).

**Positive economic effects**. When an earthquake happens, economic losses usually occur; however, one or various economic sectors can face positive effects in certain regions. Using the same indicators, we can measure the positive effects in the same components of the economy for which we calculated the negative impacts. These probabilistic indicators of positive effects can also

be helpful for better understanding and managing the seismic risk. Similar to economic losses, positive effects are captured for a vast collection of events that collectively describe the seismic hazard of the country. Our analysis confirms that, for specific events, there are gains, not losses, in some sectors and regions. For instance, the average annual gain of production computed for Chile was 18.32 million US dollars, that is, 0.0041% of its total yearly production. Figure 3 and Fig. 6 in the most-right panels, and Supplementary Fig. 7 present examples of the probabilistic indicator of positive effects obtained for Chile. By now, we are not considering recovery and reconstruction processes of the lost capital stock, the reason why, on average, positive effects are much smaller than the negative effects and mainly due to substitution effects.

**By-scenario analysis**. CPI changes, production losses, and employment losses can also be probabilistically computed for any particular event coherent with the seismic hazard model. For instance, we present the direct losses, production losses, and GRP reductions estimated for simulations of three large earthquakes occurred in Chile in 1960, 1985, and 2010 (Supplementary Fig. 8 and Supplementary Fig 9). Risk results by event can also be obtained at the regional level, as shown for the Mw8.8 event similar to the Maule 2010 Earthquake in Supplementary Fig. 10. As an example of the valuable information that we can obtain, we found that for the Maule event, the maximum price increase takes place in the Region of Biobío (R10), with a 1.9% (Supplementary Fig. 6b and Supplementary Fig. 10g) and would cause an average production loss of 1.7% of the total yearly production of Chile. Simulation of an individual scenario of indirect losses, especially those more critical, can turn out useful for holistic risk management, i.e., better design of rapid emergency response plans, better post and pre disaster economic strategies, and others.

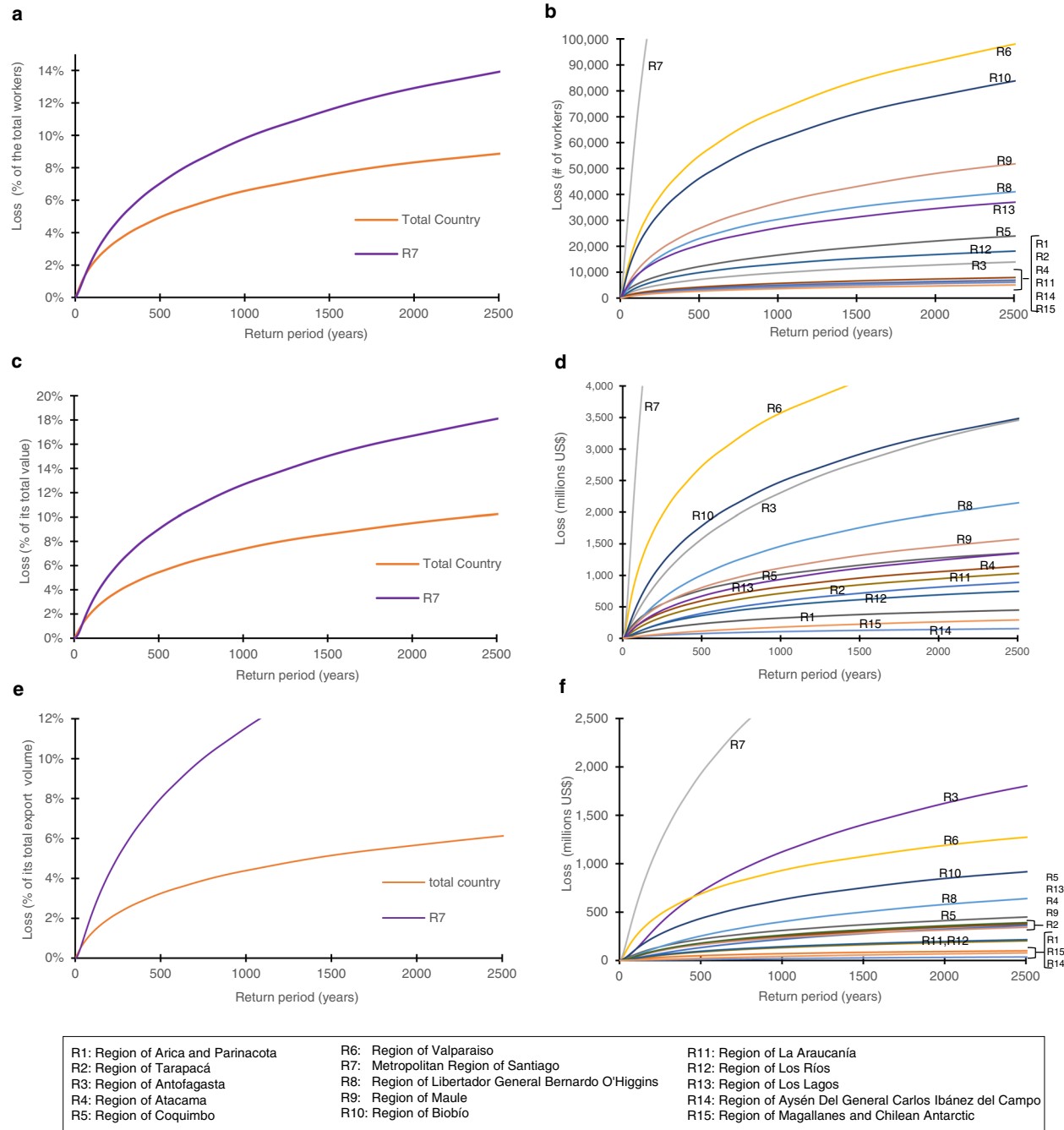

**Fig. 5 Loss Exceedance Curves (LECs) for Chile of employment, Gross Domestic Product (GDP), Gross Region Product (GRP), and export volume.**
**a** Shows the loss exceedance curves of employment for the whole country and Region 7 (Metropolitan Region of Santiago). **b** Shows the employment loss exceedance curves for each region of Chile. **c** Illustrates the loss exceedance curves of GDP for Chile and the loss exceedance curve of GRP for Region 7. **d** Illustrates the GRP loss exceedance curves for each region of Chile. **e** Shows the loss exceedance curves of export volume for the whole country and Region 7. **f** Shows the loss exceedance curves of export volume for each region of Chile.

By-scenario loss analysis is also helpful in calibrating and partially validating catastrophe models when information of real losses of recent events is available. In our case, official information of direct and indirect losses of the 2010 Maule Earthquake is available. However, empirical validation of catastrophe models is complex. We will address this issue in more detail in the discussion section.

According to estimations of the Central Bank of Chile, The Maule 2010 Earthquake caused 3% of losses in the total net capital stock of the Chilean economy, 3.2% in the residential buildings, and 2.6% in non-residential infrastructure[43]. Our model estimates direct losses in non-residential buildings of 2.5% of their total value. Furthermore, the Chilean Government[44] estimated a GDP decrease of 7600 million dollars for the next 4 years after the 2010 Maule Earthquake. Our simulation for this event estimates a yearly GDP contraction of 1.65%, which is coherent with the official estimation, assuming that the first year after the earthquake, at least half of the total 7600 million loss

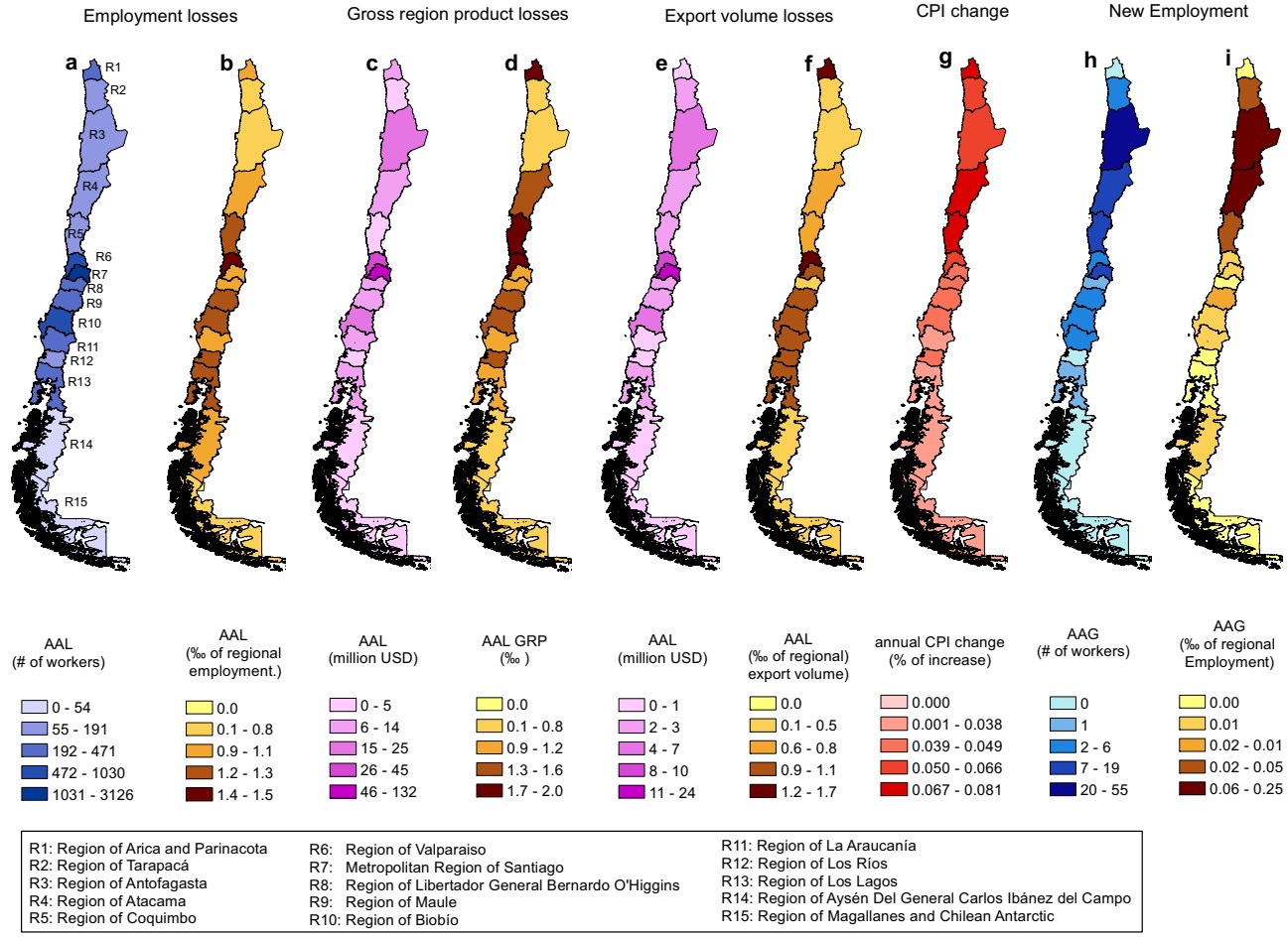

**Fig. 6 New risk indicator for Chile. a**, **b** Show the Average Annual Loss (AAL) of employment by region of Chile in number of workers and as a fraction (per thousand) of the corresponding regional employment. **c**, **d** Show the AAL of Gross Regional Product (GDP) in millions of dollars and relative terms. **e**, **f** Show the AAL of export volume by region in millions of dollars and as a fraction (per thousand) of the regional export volume. **g** Shows the average annual change of the Consumer Price Index (CPI) by region of Chile. **h**, **i** Show the Average Annual Gain (AAG) of employment by region in number of workers and as a fraction (per thousand) of the regional employment.

took place, that is, a 1.74% GDP contraction (Total Chilean GDP 2010: 218,500 million dollars).

## Discussion

Our approach has shown how the CGE model and the probabilistic model for seismic risk assessment can work together, allowing for a robust and systematic probabilistic glance of the consequences of the earthquake disturbance into the whole economy. We were able to successfully deal with the production losses, exemplifying our methodology with the computation of the standard risk metrics concerning production losses, that is, the average annual production loss and the production loss exceedance curve, both at country and regional levels, and computed by considering a vast collection of earthquakes, with known annual occurrence frequencies, that collectively describe the entire seismic hazard of a country. We have also shown how our approach can go beyond production losses, harnessing the richness of the CGE modeling. We have computed the standard risk metrics for different economic variables, such as losses of employment, GDP, GRP, export volume, and CPI. We believe that these risk measures are useful and complement the usual risk indicators of physical damage and contribute to a better integral disaster risk management and design of financial hedging instruments for governments and the insurance industry. Our model captures the negative consequences of earthquakes and

some positive effects on the economy, and we propose probabilistic measures of these gains. Our methodology is not exclusive to earthquakes and can be easily extended to other natural disasters like flooding, hurricanes, and droughts. The main results show that disaster management measures and damage prevention activities must consider intersectoral and interregional linkages in the economy. Supply-chain disruptions are propagation channels of the extreme event impacts, which is directly related to regional vulnerability to such events.

The modeling approach is fully probabilistic on the earthquake occurrence side, but for now, it is deterministic on the CGE side; the reason for this is that, while earthquake risk modelling is a 50-year-old discipline, within which uncertainties have been thoroughly studied and characterized, we feel that characterization of uncertainties in CGE modelling is still a work in progress.

The possibility to aggregate and transfer the macro-economic assessment of disaster impacts to regional levels of adaptation and analysis is limited[24]. However, regional modeling is essential as an exercise to measure the systemic effects of disruptions in supply chains, whether caused by natural hazards, extreme events, or disasters. The regional evaluation of the wider indirect economic impacts of natural disasters and climate change still requires new analytical tools. Therefore, as done in our analysis, mapping the economic impact of the natural disaster is essential for risk management and prevention since it provides tools for spatial

planning decisions. In this context, our study can contribute to the literature by considering the economic impacts of natural hazards at the regional level using integrated modeling based on the probabilistic risk model and the spatial CGE model.

We proposed to use spatial CGE models (SCGE), unlike some of the previous studies that assess the impact of natural disasters from interregional input-output models. CGE models go much further than input-output models, in which the principal focus is on links through sales of goods and services between industries and from industries to final users. In contrast, CGE modeling identifies behavior by individual agents and emphasizes links provided by competition for scarce resources[45]. Koks et al. (2016)[46] compared natural disaster impacts using an input-output model and a CGE model. They showed that for a detailed assessment of disaster impacts on the economy, including the price effects and effects on employment, the CGE models are better suited. In addition, they highlight that the conventional multi-regional input-output models may largely overestimate the losses.

Although our approach mainly uses SCGE models in a short-run environment, the type of CGE modeling proposed in our work allows us to easily change to a long-run economic scenario, including the labor migration effect. A wide range of aspects can be analyzed and studied using a long-run closure; however, we have limited ourselves to presenting the magnification effect that economic losses would have if no mitigation actions and recovery of the capital stocks destroyed by earthquakes took place over time. We compared some aggregated results regarding the annual average loss and loss exceedance curves of production in Chile obtained using short-run and long-run environments of economic modeling (Supplementary Fig. 11). Thereby, we emphasize the flexibility of our methodological approach to carry out short and long-term scenarios.

The human and physical systems are strongly connected; human activities influence physical processes and vice versa[24]. Thus, physical factors can influence human adaptive behavior. For instance, risk perception is higher after an earthquake and can cause a higher uptake of adaptation measures. The human-physical interactions, in general, are missing in the current economic models. This gap can guide the development of future research for extreme events modeling. A way to incorporate these interactions is to alter utility and production functions in various ways and explicitly incorporate uncertainty. However, this involves interdependence of utility functions, which are challenging to model in general, especially in a CGE framework[45]. We explicitly deal with the uncertainties of the results by using probabilistic physical damage scenarios and the occurrence likelihood of a vast collection of earthquakes.

The capacity to compute probabilistic metrics regarding positive impacts caused by earthquakes is another contribution of this work. The positive effects are related to the systemic perspective intrinsic to the productive linkages identified through the CGE model—that is, the feedback and interactions in the productive system. The spatial CGE model captures the links between different parts of the economy and models it as a set of integrated supply chains. Thereby, the capital stock losses due to an earthquake in one region result in changes in sectoral output in other regions through the spillover effects they cause across the supply chain. The effects on the supply chain are the indirect costs identified by the CGE model, that is, induced by disruption of economic activities in other, linked regions. Some regions may have positive effects (GRP growth) caused by the change in relative prices captured in the CGE model. The change in relative prices causes a change in the interregional trade flows—that is, the interregional exports and imports. This change in trade flows, in turn, will have a negative/positive impact on Gross Regional

Product. A subsequent increased economic activity because of spending on infrastructure and reconstruction can also partly offset the negative economic consequences of earthquakes[47]. This impact is part of the long-term results related to the mitigation actions and recovery of the capital stocks.

Empirical validation of catastrophe models is, by definition, a difficult task. Fortunately, catastrophes are relatively rare events, so observed values of losses are never a sample big enough to allow for empirical validation. Even if we had observed losses for a long time, cities change, construction materials change, so the loss information of events that took place more than a few decades ago is not very useful. In a way, the reason to start developing catastrophe models back in the 1990s was precisely the need to compensate for this data shortage. So direct validation of the models, in the sense of empirically establishing the exceedance frequency of losses of various sizes, is never possible.

Nevertheless, efforts are made to carry out partial validations of different kinds. First, the rate of occurrence of earthquakes of various magnitudes, as well as their spatial distribution, is estimated from appropriate earthquake catalogs and knowledge of the regional tectonic setting; this guarantees that the model of future earthquake occurrences will not be introducing too many or too few events; this also guarantees that the spatial distribution of future, hypothetical events, will be coherent with the observed distribution of real earthquakes and coherent also with geological science.

Additional validations are made regarding the relation between earthquake source characteristics (magnitude, hypocentral location, rupture plane orientation, etc.) and the ground acceleration field produced by the event. This guarantees that, on average, the observed ground accelerations and the accelerations predicted for future events will be unbiased.

In some cases, it is possible to compare the actual losses produced by an event with those computed with the model for a synthetic event of similar characteristics. We presented an example of this comparison with the Maule earthquake of 2010, finding that the modeled losses are coherent with those observed. In some cases, several loss-producing events can be used in this validation phase, but, in our case, the last big event before the Maule earthquake took place in 1985, which was considered to be too far away in the past.

Natural disasters in a given area affect assets located in the area. Nevertheless, losses can be transmitted to other areas through various channels, such as trade linkages, demand linkages, and interregional mobility of production factors[48]. A better understanding of these spatial and economic dimensions of natural disasters is crucial to properly measure human and economic losses and direct efforts to prepare for and mitigate the losses. Disaster planning and preparedness become more effective when the total cost (direct and indirect) is calculated. Although direct impacts are usually higher than indirect impacts, the latter is increasingly important for total economic loss assessment; neglecting these spillover effects may lead to poor allocation of funds during recovery[49]. Policies should consider relief actions to mitigate both the impact on directly affected areas and indirectly affect neighboring regions[50].

## Methods

**General approach**. We start generating a stochastic earthquake catalog that is coherent with a seismic hazard model for the region under study. We then simulate the occurrence of the first earthquake in the catalog and compute the physical damage (direct losses) produced to buildings, factories, and infrastructure using the conventional probabilistic seismic risk assessment techniques. Direct losses perturb the initial equilibrium of the CGE model, so we rerun it until it reaches a new equilibrium, but with different levels of outputs and prices, which allows for computation of the losses due to that particular event. The analysis is then repeated for each event belonging to the stochastic earthquake catalog. Finally, we compute

various probabilistic risk metrics, including the effects of indirect losses, using the results obtained for all earthquakes. The remainder of this section will explain our approach and each of its components with some depth.

**Seismic risk model**. First, the seismic risk model contains an exposure database[51–55] that includes all assets at risk relevant to the analysis. In our case, the relevant assets are buildings, factories, infrastructure, and, in general, all assets that provide input of some kind to the economic model. In other words, all assets whose damage might have a potential impact on the economic flows. Each asset must be identified by its location, seismic vulnerability characteristics, and, particularly relevant for our present purposes, the economic sector to which it belongs.

At random instants, with all assets intact and following a Poisson process, the economic equilibrium is perturbed by the occurrence of an earthquake with known focal characteristics (magnitude, hypocentral location, orientation of the rupture plane), which in turn will produce a spatial random field of intensities (peak ground acceleration, spectral values). In contemporary seismic risk models, this information is provided by its *hazard component*[56–59]. This component provides a potentially very large event set, each associated with an annual frequency of occurrence and one or more intensity random fields. Therefore, the hazard component provides information about how frequently different kinds of earthquakes occur and gives probabilistic indications of the intensities it produces. In principle, the hazard component should contain information about occurrence frequencies and intensity distributions of all earthquakes that could take place in the future. In other words, the event set must be collectively exhaustive.

Once a hypothetical earthquake has taken place, and its intensities are known—or, more precisely, the probability distributions of the intensities are known—,the seismic risk model[60] provides tools to assess, in probabilistic terms, the level of direct losses suffered by each one of the assets contained in the exposure database; this part of the model is usually referred to as the loss component. The level of damage sustained by an asset depends on its location, the size of the intensity and its vulnerability characteristics. Thus, once a hypothetical event has taken place, we have means to determine the probability distribution of the losses sustained by each one of the assets at risk using special functions called vulnerability functions[60–62] that we have used to characterize the seismic vulnerability of non-residential buildings belonging to the economic sectors of Chile. Supplementary Note 2 in Supplementary Material describes the characteristics and specifications of the probabilistic seismic risk model used in this work for Chile.

In general, and given the lower geographical resolution of the CGE models compared to that of the seismic risk models, a loss aggregation is required in order to sum all the losses that correspond to the same economic sector at the same economic region. Since the losses at the various assets are not fixed numerical values but correlated random variables, the aggregation process is not trivial because of the correlation among losses for the same event. Supplementary Note 3 presents the loss aggregation process used in our study.

Therefore, as it can be noticed, the seismic risk model is used in our approach to determine two essential pieces of information for each one of the members of the event set: (1) the probability distributions of the losses incurred by assets belonging to all economic sectors and regions, that is, the severity of the direct losses; and (2) the annual frequency with which that particular loss scenario takes place. We will see later how this information is used in the overall risk calculations.

So far, the use of the seismic risk model is not at all different from its classic use in risk assessment. However, we will later see how these classic risk model results are used as inputs to the economic modeling.

**Economic model**. We use the BMCH, the Chilean version of the B-MARIA model (Brazilian Multisectoral And Regional-Interregional Analysis Model), a fully operational spatial CGE model for Chile. The model uses an approach similar to[63–65] to incorporate the interregional economic structure. We use an absorption matrix as the basis to calibrate the CGE model, together with a set of elasticities borrowed from the econometric literature applied for Chile. This database allows capturing economy-wide effects through an intricate plot of input-output relations.

The current version of the BMCH model recognizes the economic structures of the 15 Chilean regions. Results are based on a bottom-up approach—i.e., national results are obtained from the aggregation of regional results. The model identifies 12 production/investment sectors in each region producing 12 commodities, a representative household in each region, regional governments and a central government, and a single foreign area that trades with each domestic region. Two local primary factors are used in the production process, according to regional endowments (capital and labor). Particular groups of equations define government finances, accumulation relations, and regional labor markets. The BMCH model qualifies as a Johansen-type model in that the solutions are obtained by solving the system of linearized equations of the model, following the Australian tradition. A typical result shows the percentage change in the set of endogenous variables after a policy is carried out, compared to their values in the absence of such policy, in a given environment. Interregional linkages play an important role in the functioning mechanisms of the model. These linkages are driven by trade relations (commodity flows) and factor mobility (capital and labor migration). Interregional trade flows are incorporated; interregional input-output relations are required to calibrate the model, and interregional trade elasticities play an important role[41]. Supplementary Note 4 in Supplementary Material presents the complete specification of the model.

When an earthquake occurs, it produces direct losses, whose probability distributions were determined with the seismic risk model described succinctly in the previous subsection. Once aggregated, the direct losses by sector and by region are entered into the CGE model as "shocks" to the capital stock component of the sector/region combination. These shocks are nothing more than exogenous capital stock reductions, usually calculated as the ratio between physical loss and the total cost of the capital stock. When entering the set of shocks to the CGE model, the equilibrium conditions of the model are lost, so we need to rerun the CGE model to reach a new equilibrium that reflects how the economy adjusts to the received shock. The new equilibrium condition is obtained with a new value-set of the endogenous variables, which are the model results.

A CGE model can be made of hundreds or even thousands of variables (exogenous and endogenous); each can provide us with a different type of result, either of economic or social interest. The richness of the CGE model in terms of the number of results is extraordinary, allowing for the possibility of developing a broad range of analyses. Initially, we will focus on the variables that quantify the total production of the industries; later, however, we will analyze other non-economic types of losses.

We will define the production loss of sector $i$ in the spatial region $j$, $Lp_{ij}$, as the difference between the production before and after the earthquake for the same sector/region. In other words, as a consequence of the decreased capital stock in specific sectors and regions hit by the earthquake, the economy attains a new equilibrium in which the production at that sector/region is smaller (or higher) after the earthquake than before. We regard this difference as a production loss, and this will be our initial measure of indirect losses, although we will later explore the use of results related to other variables.

At this initial phase of our research, the behavioral parameters and structural coefficients of the CGE model, the parameters and coefficients required to establish the relations between exogenous and endogenous economic variables, are considered deterministic. Despite this, the outputs of the CGE model— the indirect losses—are not fixed numerical values, but random variables, because some of the inputs were also random variables. The probability distributions of all relevant CGE model outputs, either at the sector/region level or for any required aggregation, are obtained during the modeling.

At this point, we can compute, for each event of the event set, probabilistic direct and indirect losses. The following section will illustrate how the most common risk measures can be obtained with the results presented so far.

**Risk measures**. The most common risk measures, both in the disaster risk management world and in the insurance sector, are: (1) the average annual loss; and (2) the loss exceedance curve, which indicates the average frequency with which given values of loss would be exceeded. We will focus only for illustration purposes on the total direct and indirect losses. The total losses are, of course, the sum of the losses for all assets, in the case of the direct losses, and for all sectors and regions, for the case of the indirect losses.

For instance, the $k$-th event of the event set produces probabilistic direct and indirect losses $Ld_k$ and $Lp_k$, respectively. Then, the corresponding average annual losses, $AAL_d$, and $AAL_p$ would be given by:

$$AAL_d = \sum_{k=1}^{Events} \mathrm{E}(Ld_k)F_k \tag{1}$$

$$AAL_p = \sum_{k=1}^{Events} \mathrm{E}(Lp_k)F_k \tag{2}$$

where E(.) denotes expected value and $F_k$ is the annual frequency of occurrence of event $k$. $AAL_d$ is a quantity routinely computed in conventional risk analyses; $AAL_p$ is introduced in this paper.

The loss exceedance curves, $v_d(.)$ and $v_p(.)$ of direct and indirect loss, respectively, are computed with the following expressions:

$$\nu_d(l) = \sum_{k=1}^{Events} \mathrm{Pr}(Ld_k > l)F_k \tag{3}$$

$$\nu_p(l) = \sum_{k=1}^{Events} \mathrm{Pr}(Lp_k > l)F_k \tag{4}$$

where $Pr(Ld_k > l)$ and $Pr(Lp_k > l)$ are, respectively, the probabilities of direct and indirect losses exceeding a given value, $l$. In this case, $v_p(.)$ is introduced in this paper.

**Complementary risk indicators**. The procedure for the computation of the probabilistic risk measures regarding employment, GRP, CPI, export volume, and any other economic variable of the CGE model is the same but for the corresponding economic variable instead of that of production (indirect loss) one. In the case of positive economic effects, the procedure for the computation is again the same, with the difference that, in this case, we only account for the positive effects that each earthquake triggers in the CGE model.

The risk calculations shown in this paper have been carried out with program DIRAS-2020, developed at the Instituto de Ingeniería, UNAM. This software has

been specifically created to join and process the information coming from a conventional seismic risk model and a spatial CGE model.

**Reporting summary**. Further information on research design is available in the Nature Research Reporting Summary linked to this article.

## Data availability

All data generated and used in this analysis regarding to the seismic hazard model, vulnerability functions, the exposure database and the interregional general equilibrium model for Chile (BMCH) can be freely accessed at https://github.com/JALeonTorres/RAPELE-. The exposure database used data from LandScan (2017 version), available at the LandScan portal (https://landscan.ornl.gov/), the GHS-POP dataset (2015 version), available from (https://ghsl.jrc.ec.europa.eu/), the WorldPop dataset (2015 version), available from (https://www.worldpop.org/) and the nighttime scenes from VIIRS sensor (version 1, updated March 2017), available at (https://earthdata.nasa.gov/earth-observation-data/near-real-time/download-nrt-data/viirs-nrt). The source data underlying Figs. 1–5, Table 1 and Supplementary Figs. 1–11 are provided as a Source Data file.

## Code availability

The risk calculations presented in this study were performed by means of the program DIRAS-2020, freely available through https://github.com/JALeonTorres/RAPELE-. This repository includes a hands-on guideline to reproduce all results presented in this analysis. DIRAS interacts with the program CRunGEM, which is an environment for running CGE models built with the software GEMPACK. CRunGEM - version 2015 can be downloaded from (https://www.copsmodels.com/crungem.htm) and GEMPACK version 12.1 from (https://www.copsmodels.com/gpeidl.htm). For reproducing the results, it is just required a temporary license available with the trial version of GEMPACK v12.1.

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

## Acknowledgements

J.A.L was supported by the scholarship program of graduate studies from the National Council for Science and Technology of Mexico, CONACYT, and the Instituto de Ingeniería of the Universidad Nacional Autónoma de México. E.H. acknowledges financial support from CNPq (Grant 302861/2018-1), FAPESP (Grant 2018/08337-8), and the National Institute of Science and Technology for Climate Change Phase 2 under CNPq Grant 465501/2014-1 and FAPESP Grant 2014/50848-9. I.F.A. acknowledges financial support from FAPESP (Grant 2019/00057-9).

## Author contributions

J.A.L and M.O conceived and designed the conceptual framework for the study. J.A.L implemented the study and performed all calculations, simulations, and data analysis. E.H designed, and E.H and I.F.A. implemented the BMCH model for Chile. All authors contributed to writing the paper. All authors discussed the results and contributed to paper refinement.

## Competing interests

The authors declare no competing interests.
