## [Peer Review File · Nature Communications]

Risk caused by the propagation of earthquake losses through the economyReviewers' Comments:

Reviewer #1:

Remarks to the Author:

Dear authors,

thanks for your interesting article that has included elaborate SCGE analysis with many simulation runs and uses detailed information about possible future disasters. I am however missing a number of important points in your paper:

- 1) There is a need for better explanation of the difference between modelling of events with low/long return periods. It is not clear to the reader whether you treat them differently in your SCGE modelling and what are the main differences.
- 2) Your paper attempts to make the SCGE modelling for the very long term (100 000 years) without properly explaining if you assume any change in the population or economic structure in this very long-run.
- 3) It seems from the current text of the paper that there are no assumptions made about the economic/regional/demographic developments in the very long run and that you only concentrate on the indirect changes via supply chains. In that case it would be good to explain what is the value added of SCGE model as compared to MRIO analysis that is also used in the literature to assess indirect impacts of disasters.
- 4) You explain that besides negative effects disasters may also have positive effects for some regions and sectors due to substitution effects along the supply chain. However, the largest positive effects are associated with the recovery process and reconstruction of the lost capital stocks. You do not include recovery process in your modelling. What is the motivation for this?
- 5) The main value added of SCGE models is that they are able to capture the changes in regional and sectoral structure that follow some specific shocks. One would expect that repetition of extreme events would somehow change the allocation of economic activity and population and one would expect that the impacts would be larger in case of frequent events. Could you be more explicit about whether you use comparative static SCGE or recursive dynamic SCGE model and also add a discussion about dynamic effects into the paper.

Finally, maybe you would like to have a look at the results of the large European research project that is doing quite similar research by developing general methodology for evaluation of indirect effects of extreme events <https://www.espon.eu/natural-disasters>.

Reviewer #2:

None

Reviewer #3:

Remarks to the Author:

The manuscript proposes an approach to combine a probabilistic seismic risk model with an economic model to forecast direct and indirect economic costs of earthquakes for Chile. It is written that the main novelty is the combination of these two models and that the model results are illustrated with the case of Chile. However, in the main text, main emphasis is on the Chilean results and the combination and approach is not that much detailed. It could thus be explained more carefully that the approach in the manuscript is exclusively on Chile, and Chile could also be included in the title. It is also written in the main text, that the approach is merely a proof-of-concept instead of being a final product, as all data could not be collected. However, being a proof-of-concept, the approach could be detailed, justified and discussed more, as currently the focus is on the results. Additionally, the results of the manuscript are not groundbreaking as such, and they lack a proper validation. It is discussed that the results are feasible and robust but it is not properly justified why it is so. Therefore, some

kind of validation, e.g. in relation to historical data could have made a good extension for the manuscript and increase its robustness. Despite these flaws, I think that the approach is novel and interesting but I am unsure whether it is sufficiently novel.

In my opinion, the seismic risk model is not presented in a sufficient detail currently and its quality is difficult to assess. Full model description (similar to that of economic model) should be given in the supplementary material. In particular, methods, maps and/or results of the hazard model should be given and it should be detailed more specifically where do the information about loss components come from.

Discussion section of the manuscript is currently very short and the relevancy and importance of findings could be discussed much more carefully. In particular, how do you know that your results are realistic? This is also visible in the abstract which does not currently include a proper conclusion, and in the final sentence of introduction (lines 112-113) in which the idea could be extended and clarified.

The results of the manuscript suffer from multiple problems. In particular, the reference to tables and figures could be given in parentheses in the end of sentences instead of writing "Figure 1 shows" etc. There are also some unnecessary parts that could be removed and/or moved to figure/table captions (e.g. lines 142-145, 166-168, 170-173). The results are currently mostly descriptive and could be of interest for some, but for a wider audience, they could be written more succinctly, and could also be more in-depth. One option is to conduct a proper validation, as I already wrote above. Another option is to provide some kind of uncertainty estimates or confidence intervals. It is written that the economic model is deterministic but this choice is not justified. Tables and figures could also be reworked a bit; for instance, table 1 would need a better caption (are main results in the AAL column?) and there could be an additional volume showing the AAL in percentages of total value. Then, sector and region names should be explained somewhere in main text, as currently e.g. Figure 2 is difficult to interpret. Finally, "per mile" is written several times in results. This should be translated into English as mile is a unit of length in English.

Although the language seems to be mostly fine, there are some problems with it. For instance, in abstract line 6, "catastrophe-induced" seems to be unnecessary, and in line 12 word "respectively" is missing. There are also problems in main text (e.g. end of sentence in line 44, awkward phrasing of the approach in lines 63-65, missing references in line 74, unnecessary use of "etc." in line 82, somewhat normative statement of Chilean economy in lines 96-98, "C" and "being" being in wrong order in line 152 and so on).

Response to reviewer comments.

Reviewer #1

Dear authors,

thanks for your interesting article that has included elaborate SCGE analysis with many simulations run and uses detailed information about possible future disasters. I am however missing a number of important points in your paper:

- 1) There is a need for better explanation of the difference between modelling of events with low/long return periods. It is not clear to the reader whether you treat them differently in your SCGE modelling and what are the main differences.

Action taken: There is no difference in the way we model events with short or long return periods. As we will show while answering the following remark, an event with a long return period is not an event that will occur far away in the future, but just an event with a low probability of occurring in the next year. In this sense, large and small earthquakes are modelled using the same techniques, but the small events are more likely to occur –or more frequent- than the large ones.

- 2) Your paper attempts to make the SCGE modelling for the very long term (100 000 years) without properly explaining if you assume any change in the population or economic structure in this very long-run.

Action taken: we addressed this point with two actions: i) We incorporated a short paragraph in the main paper: 2) we aggregated a supplementary note in Supplementary Material.

Main paper: “Note that in catastrophic risk modeling, it is customary to indicate the likelihood of an event taking place in, say, the next year, by using its return period. (See Supplementary Note 1 in Supplementary Material).” – Page 6.

Supplementary Note: “In catastrophic risk modeling, it is customary to indicate the likelihood of an event taking place in, say, the next year, by using its return period. In this context, the return period of any given loss value, l , is the average time between events that produce losses equal or greater than l . Therefore, if a loss value has a return period of 100,000 years, this means that events that produce losses greater or equal than the given value occur, on average, every 100,000 years. But this does not mean that the event will take place 100,000 years in the future from now. It just means that the likelihood of this event taking place in the next year is very low. How low? Since we are considering a Poisson occurrence process in time, the annual probability of exceedance, Pe , and the return period, T are related through:

$$Pe = 1 - e^{-\frac{1}{T}}$$

For large values of T ,

$$Pe = \frac{1}{T}$$

Therefore, events with large return periods are not events that will occur far away in the future; they are simply unlikely events. There is, thus, no need to account for changes in population or economic structures. Another heuristic way to see our simulation scheme is thinking that, in reality, we are not simulating the next 100,000 years, but 100,000 times the following year.” – Page 8, Supplementary Material.

- 3) It seems from the current text of the paper that there are no assumptions made about the economic/regional/demographic developments in the very long run and that you only concentrate on the indirect changes via supply chains. In that case it would be good to explain what is the value added of SCGE model as compared to MRIO analysis that is also used in the literature to assess indirect impacts of disasters.

Action taken: We addressed these points in the following way: (i) we highlight the advantages of using the SCGE model as compared to MRIO analysis to assess the indirect impacts of disasters; (ii) we run the simulations in the long-term scenario and included those results as a comparison with the original ones (short-run environment).

The paragraphs that we incorporated in the new version are:

“We use a spatial CGE model, unlike a part of previous studies that assess the impact of natural disasters from interregional input-output models. CGE models go much further than the input-output model, in which the principal focus is on links through sales of goods and services between industries and from industries to final users. In contrast, CGE modeling identifies behavior by individual agents and emphasizes links provided by competition for scarce resources (Dixon et al., 2017). Koks et al. (2016) compared natural disaster impacts using an input-output model and a CGE model. They showed that for a detailed assessment of disaster impacts on the economy, including the price effects and effects on employment, the CGE models are better suited. In addition, they highlight that the conventional multiregional input-output models may largely overestimate the losses.” – Page 15

“Although our approach mainly uses SCGE models in a short-run environmental, the type of CGE modeling proposed in our work allow us easily changing to a long-run economic scenario, including between other, the labor migration effect. A wide range of aspects can be analyzed and studied using a long-run model, however, we have limited to present the magnification effect that economic losses would have if no mitigation actions and recovery of the capital stocks destroyed by earthquakes took place along the time. We compared some aggregated results regarding the annual average loss and loss exceedance curves of production in Chile obtained using a short-

run and a long-run environment of economic modeling (Supplementary Fig. 11). We would like to highlight the potential of our approach to also carry out probabilistic analysis of economic impacts at long-run.” – Page 15.

Dixon, P. B., Rimmer, M. T., Wittwer, G., Rose, A., & Heatwole, N. (2017). *Economic Consequences of Terrorism and Natural Disasters: The Computable General Equilibrium Approach*. In Abbas, A. E., Tambe, M., & Von Winterfeldt, D. (Eds.). *Improving Homeland Security Decisions*. Cambridge University Press., p. 158-192.

Koks, E. E., Carrera, L., Jonkeren, O., Aerts, J. C., Husby, T. G., Thissen, M. & Mysiak, J. (2016). Regional Disaster Impact Analysis: Comparing Input–Output and Computable General Equilibrium Models. *Natural Hazards and Earth System Sciences*, 16(8), 1911-1924.

- 4) You explain that besides negative effects disasters may also have positive effects for some regions and sectors due to substitution effects along the supply chain. However, the largest positive effects are associated with the recovery process and reconstruction of the lost capital stocks. You do not include recovery process in you modelling. What is the motivation for this?

Action taken: To better define the interpretation of our results, we added theses sentences in the discussion section of the manuscript:

“The capacity to compute probabilistic metrics regarding to positive impacts caused by earthquakes is another contribution of this work. The positive effects are related to the systemic perspective intrinsic to the productive linkages identified through the CGE model - that is, the feedbacks and interactions that occur in the productive system. The spatial CGE model captures the links between different parts of the economy and models it as a set of integrated supply chains. Thereby, the capital stock losses due to an earthquake in one region result in changes in sectoral output in other regions through the spillover effects they cause across the supply chain. The effects on the supply chain are the indirect costs identified by the CGE model, that is, induced by disruption of economic activities in other, linked regions. Some regions may have positive effects (GRP growth) caused by the change in relative prices captured in the CGE model. The change in relative prices causes a change in the interregional trade flows – that is, the interregional exports and imports. This change in trade flow, in turn, will have a negative/positive impact on Regional Gross Product. The negative economic consequences of earthquakes are partly offset by a subsequent period of increased economic activity, due to higher spending on infrastructure and reconstruction (Seville et al., 2014). Nevertheless, recovery process modeling is not part of the scope of our study.” – Page 16.

Seville E, Vargo J, Noy I (2014) *Economic Recovery Following Earthquakes Disasters*. In: M. Beer, I. Kougioumtzoglou, E. Patelli, I K. Au (Eds.) *Encyclopedia of Earthquake Engineering*. Springer, Berlin, Heidelberg.

- 5) The main value added of SCGE models is that they are able to capture the changes in regional and sectoral structure that follow some specific shocks. One would expect that repetition of extreme events would somehow change the allocation of economic activity and population and one would expect that the impacts would be larger in case of frequent event. Could you be more explicit about whether you use comparative static SCGE or recursive dynamic SCGE model and also add a discussion about dynamic effects into the paper.

Action taken: Dynamic general equilibrium models are helpful for modeling the process of capital accumulation in the economy and used to generate forecasts for industries, labor force groups, and regions; these models would not be useful for the hazard assessment caused by the human-physical interactions. To clarify this issue, we add this paragraph to the manuscript:

“The human and physical systems are largely connected; human activities influence physical processes and vice versa (European Commission, 2020). Thus, physical factors can influence human adaptive behavior. For instance, risk perception is higher after an earthquake and can cause a higher uptake of adaptation measures. The human-physical interactions, in general, are missing in the current economic models. This gap can guide the development of future research for extreme events modeling. A way to incorporate these interactions is to alter utility and production functions in various ways and explicitly incorporate uncertainty. However, this involves interdependence of utility functions, which are difficult to model in general, especially in a CGE framework (Dixon et al., 2017). The uncertainty also can incorporate into modeling as done in our methodological approach. Our scenarios understand a range of impacts that disasters can cause in the high-frequency and lower-impact events or the higher-impact and lower-frequency events, considering a loss corresponding to an average loss over a given period, which is obtained from the probabilistic analysis that considers the likelihood of occurs events.” – Page 16.

Dixon, P. B., Rimmer, M. T., Wittwer, G., Rose, A., & Heatwole, N. (2017). *Economic Consequences of Terrorism and Natural Disasters: The Computable General Equilibrium Approach*. In Abbas, A. E., Tambe, M., & Von Winterfeldt, D. (Eds.). *Improving Homeland Security Decisions*. Cambridge University Press., p. 158-192.

European Commission (2020). *Comprehensive Desk Review: Climate Adaptation Models and Tools*. Study on Adaptation Modelling. CLIMA/A.3/ETU/2018/0010. Final Report.

- 6) Finally, may be you would like to have a look at the results of the large European research project that is doing quite similar research by developing general methodology for evaluation of indirect effects of extreme events <https://www.espon.eu/natural-disasters>.

Action taken: Thank you. We have revised this literature. We also included the references. Now it reads:

“ESPON-TITAN (2020 and 2021) provide evidence on the direct and indirect economic impacts of natural hazards, extreme events and disasters identifying trends, and territorial vulnerability patterns across European regions and cities. This research project uses the multi-regional input-output models to assess the extreme event's costs on the supply chain. European Commission (2020) and World Bank (2021) also use the input-output models to assess direct and indirect economic impacts of disasters in Europe.” – Page 2-3.

ESPON-TITAN. *ESPON-TITAN Territorial Impacts of Natural Disasters*. Applied Research: Interim Report. ESPON 2020 Cooperation Programme (2020).

ESPON-TITAN. *ESPON-TITAN Territorial Impacts of Natural Disasters*. Applied Research: Annex 5 Synthesis of Case Studies. ESPON 2020 Cooperation Programme (2021).

European Commission. *Comprehensive Desk Review: Climate Adaptation Models and Tools*. Study on Adaptation Modelling. CLIMA/A.3/ETU/2018/0010. Final Report (2020).

World Bank. *Investment in Disaster Risk Management in Europe Makes Economic Sense*. Economics for Disaster Prevention and Preparedness. International Bank for Reconstruction and Development / The World Bank (2021).

Reviewer #2

Regarding reviewer #2's point about moving the methods to before the Results section, unfortunately our style is that the methods sit after the discussion.

Reviewer #3

- 1) The manuscript proposes an approach to combine a probabilistic seismic risk model with an economic model to forecast direct and indirect economic costs of earthquakes for Chile. It is written that the main novelty is the combination of these two models and that the model results are illustrated with the case of Chile. However, in the main text, main emphasis is on the Chilean results and the combination and approach is not that much detailed. It could thus be explained more carefully that the approach in the manuscript is exclusively on Chile, and Chile could also be included in the title.

Action taken: We thank very much for her/his comment because it touches on an important point. We agree that perhaps too much importance is being given in the paper to explaining the results at the expense of space and detail devoted to explaining the calculation approach. We have tried to achieve a better balance in the new version of the paper by emphasizing to the approach.

We have emphasized our approach by expanding the discussion section, writing the results section more succinctly, and integrating some modifications in the abstract and introduction section, as you can see in the paragraphs highlighted through the manuscript.

In the new version of the manuscript, we address the issues about the methodology and treatment of data regarding the choice of assumptions, validation and seismic risk techniques. In addition, we improve the discussion about the modeling, concept, and results, place our work in the context of the broader literature, and explore its implications. In particular, we show how our findings can be applied in the context of other economies.

Having said that, we hope that it is now clearer that the approach is general and that we use the case of Chile just as an example. Given this, we have deemed unnecessary it to include reference to Chile in the title.

- 2) It is also written in the main text, that the approach is merely a proof-of-concept instead of being a final product, as all data could not be collected. However, being a proof-of-concept, the approach could be detailed, justified and discussed more, as currently the focus is on the results.

Action taken: The reviewer is correct. As indicated in the previous response, we have tried to change the balance of the paper, giving more emphasis to describing the approach and less on describing the results. All modifications done are highlighted throughout the manuscript.

- 3) It is discussed that the results are feasible and robust but it is not properly justified why it is so. Therefore, some kind of validation, e.g. in relation to

historical data could have made a good extension for the manuscript and increase its robustness

Action taken: The reviewer raises a good point, to which we dedicated just a few lines in the original version of the manuscript. To address this point, we have included the following paragraphs in the main paper:

“Empirical validation of catastrophe models is, by definition, a difficult task. Fortunately, catastrophes are relatively rare events so observed values of losses are never a sample big enough to allow for empirical validation. Moreover, even if we had observed losses for a long time, cities change, construction materials change as well, so the information loss about events that took place more than a few decades ago is not very useful. In a way, the reason to start developing catastrophe models, back in the 1990s, was precisely the need to compensate for this data shortage. So direct validation of the models, in the sense of empirically establishing the exceedance frequency of losses of various sizes, is never possible.

Nevertheless, efforts are made to carry out partial validations of different kinds. First, the rate of occurrence of earthquakes of various magnitudes and their spatial distribution are estimated from appropriate earthquake catalogs and knowledge of the regional tectonic setting. This guarantees that the model for future occurrences will not introduce too many or too few events and that the spatial distribution of future, hypothetical events, will be coherent with the observed distribution and coherent with geological science.

Additional validations are made regarding the relation between earthquake source characteristics (magnitude, hypocentral location, rupture plane orientation, etc.) and the ground acceleration field produced by the event. These guarantees that, on average, the observed ground accelerations and the accelerations predicted for future events will be unbiased.

In some cases, it is possible to compare the real losses produced by an event with those computed with the model for a synthetic event of similar characteristics. We presented an example of this comparison with the Maule earthquake of 2010, finding that the modeled losses are coherent with those observed. In some cases, several loss-producing events can be used in this validation phase but, in our case, the last big event before the Maule earthquake took place in 1985, which was considered to be too far away in the past.” Page 17.

In addition, in the results section, we have aggregate simulations of the losses caused by three large earthquakes occurred in Chile in 1960, 1985, and 2010 and compared the real losses estimated by the Chilean government and those computed with our model for the Maule 2010 Earthquake.

“For instance, we present the direct losses, production losses, and GRP reductions estimated for simulations of three large earthquakes occurred in Chile in 1960, 1985, and 2010 (Supplementary Fig. 8 and Supplementary Fig 9).”
Page 13.

“According to estimations of the Central Bank of Chile, The Maule 2010 Earthquake caused a 3% of losses in the total net capital stock of Chilean economy, 3.2% in the residential buildings and 2.6% in non-residential infrastructure (Banco de Chile 2010). Our model estimates direct losses in non-residential buildings of 2.5% of their total value. Furthermore, the Chilean Government (Gobierno de Chile 2010) estimated a GDP decrease of 7,600 million dollars for the next four years after the 2010 Maule Earthquake. Our simulation for this event estimates a yearly GDP contraction of 1.65%, which is coherent with the official estimation, assuming that the first year after the earthquake, at least half of the total 7,600 million loss took place, that it is, a 1.74% GPD contraction (Total Chilean GPD 2010: 218,500 million dollars).”
Page 14.

Gobierno de Chile. Plan de Reconstrucción Terremoto y Maremoto del 27 de febrero de 2010, Resumen Ejecutivo (2010).

Banco Central de Chile. Informe de Política Monetaria – Marzo 2010 (2010).

- 4) In my opinion, the seismic risk model is not presented in a sufficient detail currently and its quality is difficult to assess. Full model description (similar to that of economic model) should be given in the supplementary material. In particular, methods, maps and/or results of the hazard model should be given and it should be detailed more specifically where do the information about loss components come from.

Action taken: We thanks this useful suggestion of the reviewer. To address this point, we have aggregated a new Supplementary Note 1 in Supplementary Material with a detailed description of the seismic risk model, dedicating an individual section to each of its components. Page 8-14.

- 5) Discussion section of the manuscript is currently very short and the relevancy and importance of findings could be discussed much more carefully. In particular, how do you know that your results are realistic? This is also visible in the abstract which does not currently include a proper conclusion, and in the final sentence of introduction (lines 112-113) in which the idea could be extended and clarified.

Action taken: We have included more extended and substantial discussions regarding the relevance of our findings. We have also included/modified the conclusion in the abstract and the introduction section.

We expanded the discussion section by adding the paragraphs presented in the previous responses and also adding the following sentences:

“Our main results show that damage prevention activities must consider sectoral and regional linkages in disaster management measures. These findings can be used with information about patterns of regional vulnerability to provide inputs for the formulation of strategies that better considering the systemic effects of natural hazards in an integrated framework of inter-regional supply chains and trade flows.” – Page 5.

“The possibility to aggregate and transfer the macro-economic assessment of disaster impacts to regional levels of adaptation and analysis is limited (European Commission, 2020). However, regional modeling is essential as an exercise to measure the systemic effects of disruptions in supply chains, whether caused by natural hazards, extreme events, or disasters. The regional evaluation of the wider indirect economic impacts of climate change still requires the development of new analysis tools. Therefore, as done in our analysis, mapping the economic impact of the natural disaster is essential for risk management and prevention since it provides tools for spatial planning decisions. In this context, our study has the potential to contribute to the literature by considering the economic impacts of natural hazards at the regional level using integrated modeling based on the probabilistic risk model and the spatial CGE model.” – Page 15.

European Commission (2020). *Comprehensive Desk Review: Climate Adaptation Models and Tools*. Study on Adaptation Modelling. CLIMA/A.3/ETU/2018/0010. Final Report.

- 6) The results of the manuscript suffer from multiple problems. In particular, the reference to tables and figures could be given in parentheses in the end of sentences instead of writing "Figure 1 shows" etc. There are also some unnecessary parts that could be removed and/or moved to figure/table captions (e.g. lines 142-145, 166-168,170-173). The results are currently mostly descriptive and could be of interest for some, but for a wider audience, they could be written more succinctly, and could also be more in-depth. One option is to conduct a proper validation, as I already wrote above. Another option is to provide some kind of uncertainty estimates or confidence intervals. It is written that the economic model is deterministic but this choice is not justified. Tables and figures could also be reworked a bit; for instance, table 1 would need a better caption (are main results in the AAL column?) and there could be an additional volume showing the AAL in percentages of total value. Then, sector and region names should be explained somewhere in main text, as currently e.g. Figure 2 is difficult to interpret. Finally, "per mile" is written several times in

results. This should be translated into English as mile is a unit of length in English.

Action taken: Thank you. We have revised this part and addressed it as follows. We have moved all references to tables and figures to the end of the corresponding sentences. We have removed/moved some parts (lines 142-145, 166-168, 170-173) from the main text to the Figures/Tables captions/description (revised manuscript lines 160-161, 193-195, 206-208, 222-223). We have reworked Table 1 with a better caption, and we included an additional column showing the AAL in the percentage of total value as suggested. We have included Table 2 in the main text with the description of the regions and sectors of our Chilean's case of study. Finally, we have replaced the word "mile" by "thousand" throughout the entire document.

We have addressed the concern of the reviewer about our model being deterministic on the CGE side. For that purpose, we have added the following lines:

"Our modeling approach is fully probabilistic on the earthquake occurrence side, but for now, it is deterministic on the CGE side; the reason for this is that, while earthquake risk modelling is a 50-year old discipline, within which uncertainties have been thoroughly studied and characterized, we feel that characterization of uncertainties in CGE modelling is still a work in progress". - Page 4.

We also have rewritten our results more briefly and trying to be more in-depth and for a wider audience. We have addressed the comments regarding the proper validation of our results above (in point 3).

- 7) Although the language seems to be mostly fine, there are some problems with it. For instance, in abstract line 6, "catastrophe-induced" seems to be unnecessary, and in line 12 word "respectively" is missing. There are also problems in main text (e.g. end of sentence in line 44, awkward phrasing of the approach in lines 63-65, missing references in line 74, unnecessary use of "etc." in line 82, somewhat normative statement of Chilean economy in lines 96-98, "C" and "being" being in wrong order in line 152 and so on).

Action taken: Thank you. We have rewritten the abstract considering the concerns about lines 6 and 12. In the main text, we have rewritten the sentences regarding concerns in lines 63-65 about awkward phrasing (revised manuscript lines 65-68). We added the missing reference in line 74 (see reviewed manuscript line 78). In line 82, we removed "etc.", and in lines (96-98), we rewrote the sentences regarding the Chilean economy (revised main text lines: 100-101). In line 152, we can modify the order or the words "C" and "being" (revised manuscript line 151). In addition, we have reviewed in deep the full text.

Reference added to line 78:

Bommer, J., Spence, R., Erdik, M. et al. Development of an earthquake loss model for Turkish catastrophe insurance. *Journal of Seismology* 6, 431–446 (2002).

Silva V., et al. Development of a global seismic risk model. *Earthquake Spectra* 36(S1) 372–394 (2020).

Dolce, M., Prota, A., Borzi, B. et al. Seismic risk assessment of residential buildings in Italy. *Bull Earthquake Eng* 19, 2999–3032 (2021).

Reviewers' Comments:

Reviewer #1:

Remarks to the Author:

Dear authors,

thanks for your revising your article in order to respond to my comments. I find your responses and changes to the paper quite satisfactory and would like to recommend your paper for publication in Nature Communications.

Reviewer #2:

Remarks to the Author:

The comments and suggestions are given in the attached PDF file.

Journal: NATURE COMMUNICATIONS

REVIEW 2 of the MANUSCRIPT entitled

“Risk caused by the propagation of earthquake losses through the economy of a country”

(ID: NCOMMS-20-40472A)

Apart from a few minor omissions pointed out by the reviewer and corrected by the authors, unfortunately, **most of the suggested corrections were not only not commented on (explained) by the authors, but also were not taken into consideration at all and implemented in the latest version of the manuscript.** It is, it must be admitted, really unexpected and unacceptable for this category of journals.

Therefore, **the authors of the manuscript are once again asked to take into consideration the comments (and provide a detailed point by point response to each one of them),** which have been suggested only for the reasons of improving the quality of the manuscript as well as its better and simpler understanding.

I. GENERAL OBSERVATIONS:

Comment 1:

In terms of the title “Risk caused by the propagation of earthquake losses through the economy of a country”, instead of the part "the economy of a country", "the economy of Chile" (or possibly, "the economy of a country on the example of Chile") would be more adequate, which more precisely states the place (location, area) of the conducted research.

Comment 2:

- The part of the text, which should represent the abstract, is not named (designated) as an abstract.
- The list of key words is missing after the abstract.
- It is noticeable that the order of the sections presented in the paper is not usual. Thus, for example, the Section METHODS (page 18) is after the Sections RESULTS (page 5) and DISCUSSION (page 14), and in fact, should instead precede them - according to the IMRAD scheme (Introduction, Methods, Results, and Discussions), which is generally accepted (by all scientific journals).

Except for the remark that the authors remained the original order of the sections (Results, Discussion, Methods), no other explanations were given regarding any of the suggestions given within Comment 2.

If the style used by the authors is in accordance with the recommendations and policy of the Journal itself, then no corrections need to be made, but the authors should have already given such an explanation in their response.

Otherwise, the suggested corrections should definitely be implemented in the manuscript.

II. DETAILED OBSERVATIONS (MAIN DOCUMENT, 30 pages):

▪ Page 1 (line 8):

Comment: The abbreviation **GDP** is not explained by its meaning (Gross Domestic Product), and conversely, the term Gross Regional Product is not given the abbreviation (**GRP**) used in the remaining part of the paper (in accordance with common practice that when some terms first appear in a paper, the full name of the term and its usual abbreviation should be given).

The authors are once again asked to correct this.

▪ Page 2 (line 26):

Comment : The title of this section is not specified (it should be "Introduction"), as it is also suggested in the Journal's "Guide to authors".

The authors are once again asked to correct this.

▪ **Page 3 (line 55):**

(NEW) COMMENT: In the added sentence "*European Commission*²⁴ and *World Bank*²⁵.....", the definite articles are missing.

▪ **Page 3 (line 78):**

(NEW) COMMENT: The added Reference (no. 34) is missing.

▪ **Page 5 (lines 115-118):**

(NEW) COMMENT: The added sentence "*These findings can be used wiper information about patterns of regional vulnerability to provide inputs for the formulation of strategies that better considering the systemic effects of natural hazards in an integrated framework of inter-regional supply chains and trade flows*" **should be improved.**

▪ **SECTION Results:**

Comment 1: (page 8, Figure 2) The figure lacks labels for the vertical axes (a meaning of percentage values and coefficients).

The authors are once again asked to correct this.

Comment 2: (page 9, Figure 3)

- For better visibility, it is suggested to increase the width of the figure to the full width of the page. The authors are once again asked to correct this.
- **(NEW) COMMENT (Figure 3 caption (added explanation) - lines 206-207):** "*AAL is presented as a percentage of its corresponding total value to see the influence of each region in the total AAL*". **Is this correct, considering the given figure and the preceeding explanation within the figure caption ("*The average annual loss in production by region is shown in panel d in millions of USD*")? Or maybe to rephrase the caption and give a more precise (and thus correct) explanation?**

Comment 3: (page 10, Figure 4) For the sake of better visibility and understanding of the given problem, it is suggested to:

- increase the height of the diagrams;
- instead of in the legend, to write the appropriate marks (R1, R2, R3, ...) next to each curve (**since there are numerous curves in the figure, as well as similar color tones that are difficult to distinguish**).

The authors are once again asked to correct this.

Comment 4: (page 11, Figure 5) For better visibility, it is suggested to:

- increase the width of the figure to the full width of the page;
- instead of in the legend, it is recommended to write the appropriate marks (R1, R2, R3, ...) next to each curve (Fig. 5, b, d, f);
- **(NEW COMMENT) the figure caption should be provided on the same page as the figure itself.**

The authors are once again asked to correct this.

(NEW) COMMENT 5: (lines 240-241) In the added sentence "*For instance, we have computed the AAL of employment, GDP, and export volume for Chile in 7,786 workers, 305 million.....(0.122% GDP contraction), and 62.....dollars, respectively*". According to Table 1 (which should also be mentioned here in the text, as the given results are presented precisely in this table), the data are not completely specified.

Comment 6: (page 12, Figure 6) For better visibility, it is suggested to increase the width of the figure to the full width of the page.

The authors are once again asked to correct this.

(NEW) COMMENT 7: (page 13, line 283) The specified Fig. 7 does not exist in the Main document.

The authors are asked to correct this.

(NEW) COMMENT 8: (page 13, lines 289-290) In the added sentence "*....we present the direct losses, production losses, and GRP reductions estimated for simulations of three large earthquakes occulted in Chile.....*", the underlined term is not correct.

▪ **SECTION Discussion:**

(NEW) COMMENT 1: (page 14, lines 306-307) In the added sentence "*Furthermore, Chilean Government.....*", the definite article is missing.

(NEW) COMMENT 2: (page 15, line 346) In the added sentence "*Although our approach mainly uses SCGE models.....*", the abbreviation **SCGE** appears for the first time in the text, and therefore, its meaning should be stated in brackets (or maybe to add the abbreviation within line 337, where the spatial CGE models are mentioned).

(NEW) COMMENT 3: (page 15, lines 352-353) The added sentence "*We want to highlight the potential of our approach also to carry out probabilistic analysis of economic impacts in the long run*" **needs to be improved.**

(NEW) COMMENT 4: (page 16, lines 361-362) The added sentence "*The uncertainty also can incorporate into modeling as done in our methodological approach*" **needs to be improved.**

(NEW) COMMENT 5: (page 16, lines 364-365) The added sentence "*...that considers the likelihood of **occurs** events...*" **needs to be improved.**

(NEW) COMMENT 6: (page 16, lines 378-379) The added sentence "*However, recovery process modeling was not part of the scope of our study but could be incorporated in future **researchers***" **needs to be improved.**

(NEW) COMMENT 7: (page 17, lines 391-393) The added sentence "*this guarantees that the model for future occurrences will not be introducing too many or too few events, and also that the spatial distribution of future, hypothetical events, will be coherent with the observed distribution and coherent also with geological science*" **needs to be improved.**

▪ **SECTION Methods:**

Comment 1: (page 19, lines 455-456) "*Since the losses at the various assets are **not numbers** but correlated random variables....*" - It would be better and more appropriate to say "*....are **not fixed numerical values**...*".

The authors are once again asked to correct this.

Comment 2: (page 20, line 468) The abbreviation **BMCH** appears for the first time in the text. Therefore, the meaning of this abbreviation should be stated in brackets.

The authors are once again asked to correct this.

Comment 3: (page 21, line 512) "*Despite this, the outputs of the CGE model – the indirect losses – are not **numbers**, but random variables, because some of the inputs were also random variables*" - It would be better and more appropriate to say "*....are **not fixed numerical values**...*".

The authors are once again asked to correct this.

III. DETAILED OBSERVATIONS (SUPPLEMENTARY INFORMATION, 26 pages):

▪ **Page 1 (Supplementary Fig. 2):**

Comment: For the sake of better visibility, it is suggested to:

- increase the height of the diagrams;
- instead of in the legend, to write the appropriate marks (R1, R2, R3, ...) next to each curve.

The authors are once again asked to correct this.

▪ **Page 3 (Supplementary Fig. 4):**

Comment: Here again, for the sake of better visibility, it is suggested to:

- increase the height of the diagrams (there is enough space to fill the whole page);
- instead of in the legend, to write the appropriate marks (S1, S2, S3, ...) next to each curve on the presented diagrams.

The authors are once again asked to correct this.

▪ **Page 4 (Supplementary Fig. 5 and Fig. 7):**

Comment: The figures lack labels for the vertical axes (a meaning of values and coefficients).

The authors are once again asked to correct this.

▪ **Page 5 (Supplementary Fig. 8):**

(NEW) COMMENT: In the added figure's caption, the part "*....the Gross Regional Product (GRP) contraction that caused the earthquake*" (lines 68-69) should be rephrased (grammatically improved), because in this form the context of the sentence is completely different (also wrong).

▪ **Page 6 (Supplementary Fig. 9):**

(NEW) COMMENT 1: In the added figure's caption, the part "*Economic losses caused in Chile three simulated earthquakes*" (line 96) should be rephrased (grammatically improved), because in this form the context of the sentence is completely different (also wrong).

(NEW) COMMENT 2: In the added figure's caption, the part "*Panel b, d, and f show....*" (line 98) should also be rephrased (grammatically improved).

(NEW) COMMENT 3: In the added figure's caption, in the part "*....the contribution of each region to both the total physical loss (purple) of non-residential buildings and the total production loss caused by the respective earthquakes.....*" (line 99), for the sake of completeness and uniformity, the remaining colour (gray) should be added in the given explanation.

▪ **Page 5 (Supplementary Fig. 8):**

(NEW) COMMENT: In the added figure's caption, the part "*....the Gross Regional Product (GRP) contraction that caused the earthquake*" (lines 68-69) should be rephrased (grammatically improved), because in this form the context of the sentence is completely different (also wrong).

▪ **Page 11 (line 230):**

(NEW) COMMENT: In the added text, the reference **Pomonis (2014)** is missing in the list of references given at the end of the supplementary material.

▪ **Page 13/14 (Supplementary Fig. 16):**

Comment:

- The horizontal axis of all diagrams is labeled "intensity (gals)". Since this parametre is not mentioned anywhere in the text, it would be good to add an explanation in the text about the meaning of this parametre and what it represents.

- **(NEW COMMENT)** The complete figure should be presented on one page of the manuscript.

The authors are once again asked to correct this.

▪ **Page 14 (line 334):**

(NEW) COMMENT: The added text "*being $E(L_j)$ and $VAR(L_j)$ the expected value*" should be grammatically improved.

▪ **Page 15:**

Comment 1: (line 351) "*Since the losses at the various assets are **not numbers** but correlated random variables....*" - As previously noted, it would be better and more appropriate to say "*....are **not fixed numerical values**....*".

The authors are once again asked to correct this.

(NEW) COMMENT 2 (line 357): Once again, the added text "*being $E(L_j)$ and $VAR(L_j)$ the expected value*" should be grammatically improved.

Comment 3: (line 367) "*Supplementary Note 2 - Specification of the **BMCH** Model*" - As previously noted, the meaning of this abbreviation should be stated in brackets.

The authors are once again asked to correct this.

▪ **Page 24/25 (Supplementary Table 3):**

Comment: The title of the table and the table itself should be given on the same page.

The authors are once again asked to correct this.

IV. REVIEWER'S RECOMMENDATION:

Taking all the abovementioned into account, the reviewer's recommendation to the editor is that **the paper be sent to the authors for revision** (in accordance with the reviewer's notes).

When the authors make suggested corrections, the reviewer's opinion is that the paper should be accepted for publication.

Reviewer #3:

Remarks to the Author:

The authors have addressed my comments from the first round and the manuscript looks good now.

Response to reviewer comments.

Reviewer #2

Apart from a few minor omissions pointed out by the reviewer and corrected by the authors, unfortunately, most of the suggested corrections were not only not commented on (explained) by the authors but also were not taken into consideration at all and implemented in the latest version of the manuscript. It is, it must be admitted, really unexpected and unacceptable for this category of journals.

Therefore, the authors of the manuscript are once again asked to take into consideration the comments (and provide a detailed point by point response to each one of them), which have been suggested only for the reasons of improving the quality of the manuscript as well as its better and simpler understanding.

Response: As kindly explained by the editor, unfortunately, in the previous decision, the comments of referee #2 were not forwarded to us, so we were not aware of these comments. Our apologies for this misunderstanding.

We appreciate the comments and suggestions from referee # 2, and below, we proceed to provide a point-by-point response.

1. GENERAL OBSERVATIONS:

Comment 1:

In terms of the title "Risk caused by the propagation of earthquake losses through the economy of a country", instead of the part "the economy of a country", "the economy of Chile" (or possibly, "the economy of a country on the example of Chile") would be more adequate, which more precisely states the place (location, area) of the conducted research.

Action taken: We agree with the suggestion and have changed the title accordingly. The new title is:

"Risk caused by the propagation of earthquake losses through the economy of a country on the example of Chile".

Comment 2:

- The part of the text, which should represent the abstract, is not named (designated) as an abstract.

Response: Unfortunately, in the format and style required by Nature Communication journal, the abstract has no title.

- The list of keywords is missing after the abstract.

Response: Unfortunately, in the format and style required by the Nature Communication Journal, there is no space in the main manuscript for the keywords.

- It is noticeable that the order of the sections presented in the paper is not usual. Thus, for example, the Section METHODS (page 18) is after the Sections RESULTS (page 5) and DISCUSSION (page 14), and in fact, should instead precede them - according to the IMRAD scheme (Introduction, Methods, Results, and Discussions), which is generally accepted (by all scientific journals).

Except for the remark that the authors remained the original order of the sections (Results, Discussion, Methods), no other explanations were given regarding any of the suggestions given within Comment 2.

If the style used by the authors is in accordance with the recommendations and policy of the Journal itself, then no corrections need to be made, but the authors should have already given such an explanation in their response.

Otherwise, the suggested corrections should definitely be implemented in the manuscript.

Response: Unfortunately, the nature communication journal style is that the methods appear after the results and discussion. The order of the section is a journal's requirement.

ii. DETAILED OBSERVATIONS (MAIN DOCUMENT, 30 pages):

Page 1 (line 8):

Comment: The abbreviation **GDP** is not explained by its meaning (Gross Domestic Product), and conversely, the term Gross Regional Product is not given the abbreviation (**GRP**) used in the remaining part of the paper (in accordance with common practice that when some terms first appear in a paper, the full name of the term and its usual abbreviation should be given).

The authors are once again asked to correct this.

Action taken: We have added the words "**Gross Regional Product**" before the abbreviation (GDP) and the abbreviation "**(GRP)**" after the words Gross Regional Product (new manuscript line 9, page 1).

Page 2 (line 26):

Comment: The title of this section is not specified (it should be "Introduction"), as it is also suggested in the Journal's "Guide to authors".

The authors are once again asked to correct this.

Action taken: We have included the section title "Introduction" at the beginning of page 2.

□ **Page 3 (line 55):**

(NEW) COMMENT: In the added sentence "*European Commission²⁴ and World Bank²⁵*", the definite articles are missing.

Response: The reviewer is correct. We have added the required articles. (New manuscript line 55,56, page 3).

Page 3 (line 78):

(NEW) COMMENT: The added Reference (no. 34) is missing.

Action taken: To deal with this issue, we modified superscripts "33,35" by "33-35" in line 78, page 3 (new version).

□ **Page 5 (lines 115-118):**

(NEW) COMMENT: The added sentence "*These findings can be used wiper information about patterns of regional vulnerability to provide inputs for the formulation of strategies that better considering the systemic effects of natural hazards in an integrated framework of inter-regional supply chains and trade flows*" should be improved.

Action taken: We changed the paragraph in the new version:

“...Our main results show that disaster management measures and damage prevention activities must consider intersectoral and interregional linkages in the economy. Supply chain disruptions are propagation channels of the extreme event impacts, which is directly related to regional vulnerability to such events.” – (new manuscript lines 114-117, Page 5).

□ **SECTION Results:**

Comment 1: (page 8, Figure 2) The figure lacks labels for the vertical axes (a meaning of percentage values and coefficients).

The authors are once again asked to correct this.

Action taken: We have added the corresponding labels to the vertical axes in Figure 2. (New manuscript, page 8).

Comment 2: (page 9, Figure 3)

– For better visibility, it is suggested to increase the width of the figure to

the full width of the page. The authors are once again asked to correct this.

- **(NEW) COMMENT** (Figure 3 caption (added explanation) - lines 206-207): "AAL is presented as a percentage of its corresponding total value to see the influence of each region in the total AAL". Is this correct, considering the given figure and the preceding explanation within the figure caption ("The average annual loss in production by region is shown in panel *d* in millions of USD")? Or maybe to rephrase the caption and give a more precise (and thus correct) explanation?

Action taken: We have attended the reviewer's suggestions as follows: 1) we have increased the width and the height of Figure 3, and we increased its resolution; 2) we have removed from the caption of Figure 3 the sentence "as a percentage of its corresponding total value to see the influence of each region in the total AAL". (New manuscript, Page 9, lines 204-206).

Comment 3: (page 10, Figure 4) For the sake of better visibility and understanding of the given problem, it is suggested to:

- increase the height of the diagrams;
- instead of in the legend, to write the appropriate marks (R1, R2, R3, ...) next to each curve (**since there are numerous curves in the figure, as well as similar color tones that are difficult to distinguish**).

The authors are once again asked to correct this.

Action taken: We have increased the height of the graphs, and we wrote the appropriate marks next to each curve as suggested by the reviewer. (New manuscript, page 10).

Comment 4: (page 11, Figure 5) For better visibility, it is suggested to:

- increase the width of the figure to the full width of the page;
- instead of in the legend, it is recommended to write the appropriate marks (R1, R2, R3, ...) next to each curve (Fig. 5, b, d, f);
- **(NEW COMMENT)** the figure caption should be provided on the same page as the figure itself. The authors are once again asked to correct this.

(NEW) COMMENT 5: (lines 240-241) In the added sentence "For instance, we have computed the AAL of employment, GDP, and export volume for Chile in 7,786 workers, 305 million (0.122% GDP contraction), and 62 dollars, respectively". According to Table 1 (which should also be mentioned here in the text, as the given results are presented precisely in this table), the data are not completely specified.

Action taken: We have increased the graphs' width and height, and we wrote the appropriate marks next to each curve as suggested by the reviewer. We have provided the caption on the same page where Figure 5 is located. (New manuscript, page 12).

Regarding new comment 5, as suggested by the reviewer, we have included the words “as presented in Table 1” at the beginning of the paragraph to make reference Table 1. (New version, page 11, line 236).

Comment 6: (page 12, Figure 6) For better visibility, it is suggested to increase the width of the figure to the full width of the page.

The authors are once again asked to correct this.

Action taken: We have increased the width of the Figure 6 to the full width of the page. In addition, we increased a little bit the height of the Figure for a better visibility. (New manuscript, page 13).

(NEW) COMMENT 7: (page 13, line 283) The specified Fig. 7 does not exist in the Main document.

The authors are asked to correct this.

Action taken: Thank you for this observation. It was a mistake. We have changed “Fig. 7” by “Fig. 6”. (New manuscript line 280, page 13)

(NEW) COMMENT 8: (page 13, lines 289-290) In the added sentence “...we present the direct losses, production losses, and GRP reductions estimated for simulations of three large earthquakes occulted in Chile”, the underlined term is not correct.

Action taken: We have replaced the word “acculted” by “occurred”. (New manuscript line 287, page 14)

□ **SECTION Discussion:**

(NEW) COMMENT 1: (page 14, lines 306-307) In the added sentence "*Furthermore, Chilean Government* ", the definite article is missing.

Response: We thank the reviewer for the observation. We have added the missing article. (New manuscript, line 303, page 13).

(NEW) COMMENT 2: (page 15, line 346) In the added sentence "*Although our approach mainly uses SCGE models.....*", the abbreviation **SCGE** appears for the first time in the text, and therefore, its meaning should be stated in brackets (or maybe to add the abbreviation within line 337, where the spatial CGE models are mentioned).

Action taken: As suggested by the reviewer, we have added the abbreviation "**SCGE**" in line 334 (page 16, new version) where the spatial CGE models are mentioned".

(NEW) COMMENT 3: (page 15, lines 352-353) The added sentence "*We want to highlight the potential of our approach also to carry out probabilistic analysis of economic impacts in the long run*" needs to be improved.

Action taken: To address this point, we replaced this sentence:

"...We compared some aggregated results regarding the annual average loss and loss exceedance curves of production in Chile obtained using short-run and long-run environments of economic modeling (Supplementary Fig. 11). Thereby, we emphasize the flexibility of our methodological approach to carry out short and long-term scenarios." – Page 15, line 350,351 new version.

(NEW) COMMENT 4: (page 16, lines 361-362) The added sentence "*The uncertainty also can incorporate into modeling as done in our methodological approach*" needs to be improved. **(NEW) COMMENT 5:** (page 16, lines 364-365) The added sentence "*that considers the likelihood of **occurs** events*" needs to be improved.

Action taken: To better define the interpretation of our results, we rewrote this paragraph in the discussion section of the manuscript:

"...A way to incorporate these interactions is to alter utility and production functions in various ways and explicitly incorporate uncertainty. However, this involves interdependence of utility functions, which are challenging to model, especially in a CGE framework⁴⁵. We explicitly deal with the uncertainties of the results by using probabilistic physical damage scenarios and the occurrence likelihood of a vast collection of earthquakes." – Page 17, lines 258-259.

(NEW) COMMENT 6: (page 16, lines 378-379) The added sentence "*However, recovery process modeling was not part of the scope of our study but could be incorporated in future researchers*" needs to be improved.

Action taken: We removed this sentence from the text. In the new version, we add this sentence:

“A subsequent increased economic activity because of spending on infrastructure and reconstruction can also partly offset the negative economic consequences of earthquakes⁴⁷. This impact is part of the long-term results related to the mitigation actions and recovery of the capital stocks.” – Page 17, lines 370-373.

(NEW) COMMENT 7: (page 17, lines 391-393) The added sentence "*this guarantees that the model for future occurrences will not be introducing too many or too few events, and also that the spatial distribution of future, hypothetical events, will be coherent with the observed distribution and coherent also with geological science*" needs to be improved.

Action taken: We have rephrased the paragraph as follows: “*this guarantees that the model of future earthquake occurrences will not be introducing too many or too few events; this also guarantees that the spatial distribution of future, hypothetical events, will be coherent with the observed distribution of real earthquakes and coherent also with geological science*”. Page 18, lines 385-388

□ **SECTION Methods:**

Comment 1: (page 19, lines 455-456) "*Since the losses at the various assets are not numbers but correlated random variables...*" - It would be better and more appropriate to say "...are **not fixed numerical values**". The authors are once again asked to correct this.

Action taken: We have replaced the word “numbers” by “fixed numerical values” (line 451 – new version, page 20):

Comment 2: (page 20, line 468) The abbreviation **BMCH** appears for the first time in the text. Therefore, the meaning of this abbreviation should be stated in brackets.

The authors are once again asked to correct this.

Response and action taken: the BMCH is the Chilean version of the B-MARIA model (Brazilian Multisectoral and Regional-Interregional Analysis Model). In the main manuscript, we have rewritten this sentence as “*We use the BMCH, the Chilean version of the B-MARIA model (Brazilian Multisectoral and Regional-Interregional Analysis Model)*”, (Page 21 -new version, line 464,465).

Comment 3: (page 21, line 512) " *Despite this, the outputs of the CGE model – the indirect losses – are not **numbers**, but random variables, because some of the inputs were also random variables*" - It would be better and more appropriate to say "...are **not fixed numerical values** ".

The authors are once again asked to correct this.

Action taken: We have replaced the work “numbers” by “fixed numerical values” (lines 509 –510 new version):

III. DETAILED OBSERVATIONS (SUPPLEMENTARY INFORMATION, 26 pages):

Page 1 (Supplementary Fig. 2):

Comment: For the sake of better visibility, it is suggested to:

- increase the height of the diagrams;
- instead of in the legend, to write the appropriate marks (R1, R2, R3,) next to each curve.

The authors are once again asked to correct this.

Action taken: We have increased the height of the diagrams, and we have written the appropriate marks next to each curve as suggested by the reviewer.

Page 3 (Supplementary Fig. 4):

Comment: Here again, for the sake of better visibility, it is suggested to:

- increase the height of the diagrams (there is enough space to fill the whole page);
- instead of in the legend, to write the appropriate marks (S1, S2, S3,) next to each curve on the presented diagrams.

The authors are once again asked to correct this.

Action taken: We have increased the height of the diagrams, and we have written the appropriate marks next to each curve as suggested by the reviewer.

Page 4 (Supplementary Fig. 5 and Fig. 7):

Comment: The figures lack labels for the vertical axes (a meaning of values and coefficients).

The authors are once again asked to correct this.

Action taken: We have added the corresponding labels to the figures as suggested by the reviewer. In addition, we increased the height of the figures.

Page 5 (Supplementary Fig. 8):

(NEW) COMMENT: In the added figure`s caption, the part " *the Gross Regional Product (GRP) contraction that caused the earthquake*" (lines 68-69) should be rephrased (grammatically improved), because in this form the context of the sentence is completely different (also wrong).

Action taken: We have rewritten the caption of Figure 8, considering the reviewer`s suggestions. The new caption of the Figure is:

“Supplementary Fig. 8 Physical losses and Gross Regional Product (GRP) contraction obtained for three simulated earthquake scenarios in Chile. Each panel presents the PGA intensity field generated by the earthquake, the physical losses in non-residential buildings, and the Gross Regional Product (GRP) contraction that caused the corresponding earthquake. Panel a simulates the occurrence of an earthquake with similar characteristics to the 1960 Mw9.5 Valdivia Earthquake, Panel b simulates the occurrence of an earthquake with similar characteristic to the 1985 Mw8.0 Valparaiso Earthquake, and Panel c an earthquake with similar characteristics to the 2010 Mw8.8 Maule Earthquake.”

□ **Page 6 (Supplementary Fig. 9):**

(NEW) COMMENT 1: In the added figure`s caption, the part "*Economic losses caused in Chile three simulated earthquakes*" (line 96) should be rephrased (grammatically improved), because in this form the context of the sentence is completely different (also wrong).

(NEW) COMMENT 2: In the added figure`s caption, the part "*Panel b, d, and f show* " (line 98) should also be rephrased (grammatically improved).

(NEW) COMMENT 3: In the added figure`s caption, in the part " *the contribution of each region to both the total physical loss (purple) of non-residential buildings and the total production loss caused by the respective earthquakes* " (line 99), for the sake of completeness and uniformity, the remaining colour (gray) should be added in the given explanation.

Action taken: We have rewritten the caption of Figure 8 taking in account the reviewer`s suggestions. The new caption of the Figure is:

“Supplementary Fig. 9 Economic losses obtained for three simulated earthquake scenarios in Chile. Panels a, c, and e show the production and physical losses caused by the earthquakes as a percentage of their corresponding total regional productions (gray bars) and their regional values of non-residential buildings (orange bars). Panels b, d, and f show the contribution of each region to the total production loss (grey bars) and the contribution of each region to the total physical loss of non-residential buildings (purple bars) caused by the earthquakes. Panels a y b present the results given the occurrence of an earthquake with similar characteristics than the 1960 Mw9.5 Valdivia Earthquake, Panels c and d present the results given the occurrence of an earthquake with similar characteristic than the 1985 Mw8.0 Valparaiso Earthquake, and Panel e and f present the loss results given the occurrence of an earthquake with similar characteristic than the 2010 Mw8.8 Maule Earthquake”.

□ **Page 5 (Supplementary Fig. 8):**

(NEW) COMMENT: In the added figure's caption, the part "*the Gross Regional Product (GRP) contraction that caused the earthquake*" (lines 68-69) should be rephrased (grammatically improved), because in this form the context of the sentence is completely different (also wrong).

Response: Thank you for your comments. We have rewritten the caption of Figure 8, considering the reviewer's suggestions.

□ **Page 11 (line 230):**

(NEW) COMMENT: In the added text, the reference **Pomonis (2014)** is missing in the list of references given at the end of the supplementary material.

Action taken: We have added this missing reference to the list of references:

"Pomonis, A. Estimation of Residential Inventory and Exposure in Urban and Rural Areas for Disaster Loss Estimation. The World Bank - Social, Urban, Rural & Resilience (GSURR), Disaster Risk Management (DRM): Washington DC, USA (2014).

□ **Page 13/14 (Supplementary Fig. 16):**

Comment:

- The horizontal axis of all diagrams is labeled "intensity (gals)". Since this parameter is not mentioned anywhere in the text, it would be good to add an explanation in the text about the meaning of this parameter and what it represents.
- **(NEW COMMENT)** The complete figure should be presented on one page of the manuscript. The authors are once again asked to correct this.

Action taken: We addressed this point with two actions: i) we have rewritten the caption of Figure 16 taking in account the reviewer's suggestions. The new caption of the Figure is:

"Supplementary Fig. 16 Vulnerability functions used for Chile. They refer to physical damage of non-residential buildings that made up the capital stock of economic sectors S1-S12. In this case, the intensity is given in terms of the spectral acceleration". Seismic vulnerability functions set up relations between the intensity of the seismic motion and the loss caused in the structure when an event of such intensity takes place".

ii) We have increased the height of the graphics of Figure 16 and now we present the Figure in one whole page of the supplementary material.

Page 14 (line 334):

(NEW) COMMENT: The added text "*being* $E(L_j)$ and $VAR(L_j)$ the expected value" should be grammatically improved.

Action taken: We have replaced the sentence "*being* $E(L_j)$ and $VAR(L_j)$ the expected value and the standard deviation of the loss of the asset j caused by earthquake k " by "where $E(L_j)$ and $VAR(L_j)$ are the expected value and the standard deviation of the loss of the asset j caused by earthquake k ": (line 365, new version)

□ **Page 15:**

Comment 1: (line 351) "*Since the losses at the various assets are not numbers but correlated random variables....*" - As previously noted, it would be better and more appropriate to say "....are **not fixed numerical values**".

The authors are once again asked to correct this.

Action taken: We have replaced the words "not numbers" by "not fixed numerical values" (line 382 – new version):

(NEW) COMMENT 2 (line 357): Once again, the added text "*being* $E(L_j)$ and $VAR(L_j)$ the expected value" should be grammatically improved.

Action taken: We have replaced the sentence "*being* $E(L_j)$ and $VAR(L_j)$ the expected value and the standard deviation of the loss of the asset j belonging to the same economic sector at the same economic region" by "where $E(L_j)$ and $VAR(L_j)$ are the expected value and the standard deviation of the loss of the asset j belonging to the same economic sector at the same economic region". (Lines 290-391, new version)

Comment 3: (line 367) "*Supplementary Note 2 - Specification of the **BMCH** Model*" - As previously noted, the meaning of this abbreviation should be stated in brackets.

The authors are once again asked to correct this.

Response and action taken: The BMCH is the Chilean version of the B-MARIA model (Brazilian Multisectoral and Regional-Interregional Analysis Model). In the supplementary note 2, we have rewritten this sentence as "We use the Chilean version of the B-MARIA model (BMCH)", (line 400). In addition, we added the next footnote:

"B-MARIA is the Brazilian Multisectoral and Regional-Interregional Analysis Model".

□ **Page 24/25 (Supplementary Table 3):**

Comment: The title of the table and the table itself should be given on the same page.

The authors are once again asked to correct this.

Action taken: As suggested by the reviewer, we have moved the title of Table 3.

iv. REVIEWER'S RECOMMENDATION:

Taking all the above mentioned into account, the reviewer's recommendation to the editor is that **the paper be sent to the authors for revision** (in accordance with the reviewer's notes).

When the authors make suggested corrections, the reviewer's opinion is that the paper should be accepted for publication.

Reviewers' Comments:

Reviewer #2:

Remarks to the Author:

The authors have endeavored to incorporate the suggested corrections into the latest version of the manuscript.

The final opinion of the reviewer is that the manuscript in this corrected version can now be accepted for publication.